# Adipocyte NR1D1 dictates adipose tissue expansion during obesity

Ann Louise Hunter[1†], Charlotte E Pelekanou[1†], Nichola J Barron[1], Rebecca C Northeast[1], Magdalena Grudzien[1], Antony D Adamson[1], Polly Downton[1], Thomas Cornfield[2], Peter S Cunningham[1], Jean-Noel Billaud[3], Leanne Hodson[2], Andrew SI Loudon[1], Richard D Unwin[4], Mudassar Iqbal[5], David W Ray[2], David A Bechtold[1]*

[1]Centre for Biological Timing, Faculty of Biology, Medicine and Health, University of Manchester, Manchester, United Kingdom; [2]Oxford Centre for Diabetes, Endocrinology and Metabolism, University of Oxford, and NIHR Oxford Biomedical Research Centre, John Radcliffe Hospital, Oxford, United Kingdom; [3]Digital Insights, Qiagen, Redwood City, United States; [4]Stoller Biomarker Discovery Centre, Division of Cancer Sciences, Faculty of Biology, Medicine and Health, University of Manchester, Manchester, United Kingdom; [5]Division of Informatics, Imaging and Data Sciences, Faculty of Biology, Medicine and Health, University of Manchester, Manchester, United Kingdom

*For correspondence:
david.bechtold@manchester.ac.uk

[†]These authors contributed equally to this work

**Abstract** The circadian clock component NR1D1 (REVERBα) is considered a dominant regulator of lipid metabolism, with global *Nr1d1* deletion driving dysregulation of white adipose tissue (WAT) lipogenesis and obesity. However, a similar phenotype is not observed under adipocyte-selective deletion (*Nr1d1[Flox2-6]:Adipoq[Cre]*), and transcriptional profiling demonstrates that, under basal conditions, direct targets of NR1D1 regulation are limited, and include the circadian clock and collagen dynamics. Under high-fat diet (HFD) feeding, *Nr1d1[Flox2-6]:Adipoq[Cre]* mice do manifest profound obesity, yet without the accompanying WAT inflammation and fibrosis exhibited by controls. Integration of the WAT NR1D1 cistrome with differential gene expression reveals broad control of metabolic processes by NR1D1 which is unmasked in the obese state. Adipocyte NR1D1 does not drive an anticipatory daily rhythm in WAT lipogenesis, but rather modulates WAT activity in response to alterations in metabolic state. Importantly, NR1D1 action in adipocytes is critical to the development of obesity-related WAT pathology and insulin resistance.

## Introduction

The mammalian circadian clock directs rhythms in behaviour and physiology to coordinate our biology with predictable changes in food availability and daily alternations between fasted and fed states. In this way, profound cycles in nutrient availability and internal energy state can be managed across multiple organ systems. A central circadian clock in the suprachiasmatic nuclei (SCN) drives daily rhythms in our behaviour (e.g. sleep/wake cycles) and physiology (e.g. body temperature), and orchestrates rhythmic processes in tissue systems across the body (*Dibner et al., 2010*; *West and Bechtold, 2015*). The molecular clock mechanism is also present in most cell types. The rhythmic transcriptome that defines cells and tissues is shaped by both local tissue clock activity and input from the central clock and rhythmic systemic signals (*Guo et al., 2005*; *Hughes et al., 2012*; *Kornmann et al., 2007*; *Koronowski et al., 2019*; *Lamia et al., 2008*; *Hunter et al., 2020*). The relative importance of these intrinsic and systemic factors remains ill-defined. Mounting evidence suggests that systemic signalling (e.g. SCN and/or behaviour-driven rhythmicity) is highly dominant in

setting daily rhythms, while local clocks serve to buffer tissue/cell responses based on time of day. Nevertheless, it is clear that our rhythmic physiology and metabolic status reflects the interaction of clocks across the brain and body (*West and Bechtold, 2015*). Disturbance of this interaction, as occurs with shift work and irregular eating patterns, is increasingly recognised as a risk factor for metabolic disease and obesity (*Broussard and Van Cauter, 2016*; *Kim et al., 2020*).

Extensive work over the past 20 years has demonstrated that circadian clock function and its component factors are closely tied into energy metabolism (*Bass and Takahashi, 2010*; *Reinke and Asher, 2019*), with strong rhythmicity evident in cellular and systemic metabolic processes. Clock-metabolic coupling in peripheral tissues is adaptable, as demonstrated by classical food-entrainment studies (*Damiola et al., 2000*; *Mistlberger, 1994*), and by recent work showing that systemic perturbations such as cancer and high-fat diet (HFD) feeding can reprogramme circadian control over liver metabolism (*Dyar et al., 2018*; *Masri et al., 2016*). Plasticity therefore exists within the system, and the role of the clock in tissue and systemic responses to acute and chronic metabolic perturbation remains a critical question. The nuclear receptor NR1D1 (REVERBα) is a core clock component, and has been highlighted as a key link between the clock and metabolism. NR1D1 is a constitutive repressor, with peak expression in the latter half of the inactive phase (daytime in the nocturnal animal). In liver, NR1D1 exerts repressive control over programmes of lipogenesis by recruiting the NCOR/HDAC3 co-repressor complex to metabolic target genes, such that global loss of NR1D1 or liver-specific deletion of HDAC3 results in hepatosteatosis (*Feng et al., 2011*; *Zhang et al., 2015*); (*Zhang et al., 2016*). Importantly, we recently showed that NR1D1 regulation of hepatic metabolism is state-dependent, with minimal impact under basal conditions yet increased transcriptional influence in response to mistimed feeding (*Hunter et al., 2020*). The selective functions of NR1D1 in white adipose tissue (WAT) are not well established and remain poorly understood. Early studies implicated an essential role of *Nr1d1* in adipocyte differentiation (*Chawla and Lazar, 1993*; *Kumar et al., 2010*); however, these findings are difficult to align with in vivo evidence. Indeed, pronounced adiposity and adipocyte hypertrophy are evident in *Nr1d1*[-/-] mice, even under normal feeding conditions (*Delezie et al., 2012*; *Hand et al., 2015*); (*Zhang et al., 2015*). Daily administration of NR1D1 agonists has also been shown to reduce fat mass and WAT lipogenic gene expression in mice (*Solt et al., 2012*), although these agents do have significant off-target actions (*Dierickx et al., 2019*). Given the links between circadian disruption and obesity, and the potential of NR1D1 as a pharmacological target, we now define the role of NR1D1 in dictating WAT metabolism.

Transcriptomic and proteomic profiling of WAT in global *Nr1d1*[-/-] mice revealed an expected de-repression of lipid synthesis and storage programmes. However, in contrast, selective deletion of *Nr1d1* in adipocytes did not result in dysregulation of WAT metabolic pathways. Loss of NR1D1 activity in WAT did, however, significantly enhance adipose tissue expansion in response to HFD feeding; yet despite exaggerated obesity, adipocyte-specific knockout (KO) mice were spared the anticipated obesity-related pathology. Integration of transcriptomic data with the WAT NR1D1 cistrome demonstrates that, under basal conditions, NR1D1 activity is limited to a small set of direct target genes (enriched for extracellular matrix [ECM] processes). However, NR1D1 regulatory control broadens to include lipid and mitochondrial metabolism pathways under conditions of obesity. Our data recast the role of NR1D1 as a regulator responsive to the metabolic state of the tissue, rather than one which delivers an anticipatory daily oscillation to the WAT metabolic programme.

## Results

### Adiposity and up-regulation of WAT lipogenic pathways in *Nr1d1*[-/-] mice

We first examined the body composition of age-matched *Nr1d1* global KO (*Nr1d1*[-/-]) mice and littermate controls (wild type [WT]). In keeping with previous reports (*Delezie et al., 2012*; *Hand et al., 2015*), *Nr1d1*[-/-] mice are of similar body weight to littermate controls (*Figure 1A*), yet carry an increased proportion of fat mass (KO: 24.2 ± 3.0% of body weight; WT: 10.8 ± 1.4%; mean ± SEM, p<0.01 Student's t-test, n=12–14/group) and display adipocyte hypertrophy (*Figure 1B*), even when maintained on a standard chow diet. Metabolic phenotyping demonstrated expected day-night differences in food intake, energy expenditure, activity, and body temperature in both KO and WT

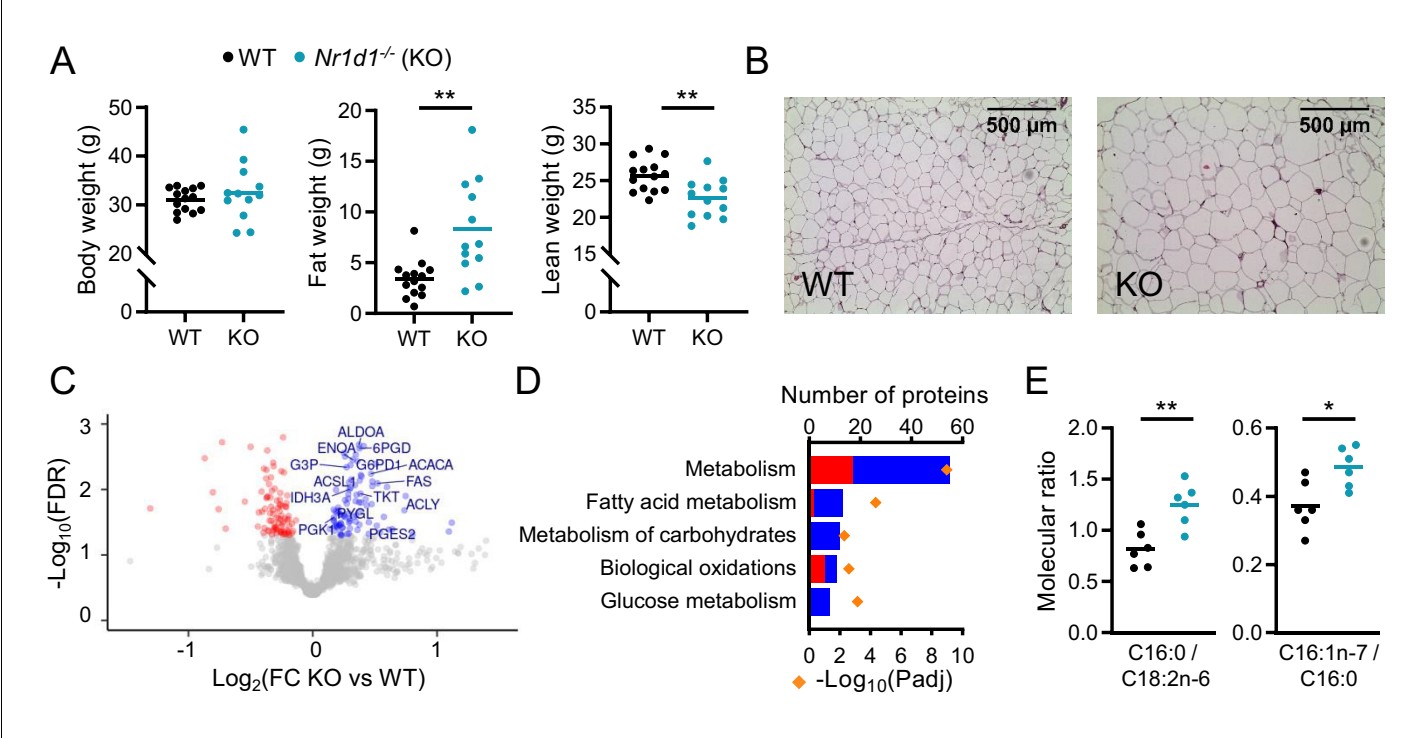

**Figure 1.** Global deletion of *Nr1d1* results in obesity and increased adipose lipid synthesis. (**A**) *Nr1d1⁻/⁻* mice exhibit significantly greater fat mass relative to wild-type (WT) littermate controls. Body weight, fat mass, and lean mass of 13-week-old males (n=12–14/group). (**B**) Increased fat mass in *Nr1d1⁻/⁻* mice is reflected in adipocyte hypertrophy in gonadal white adipose tissue (gWAT) (representative 10× H and E images shown). (**C, D**) gWAT from *Nr1d1⁻/⁻* mice exhibits a programme of increased lipid synthesis. Proteomic profiling of gWAT depots (*Nr1d1⁻/⁻* mice plotted relative to their respective weight-matched littermate controls, n=6/group (**C**)) shows deregulation of metabolic regulators and enrichment (**D**) of metabolic pathways (up- and down-regulated proteins shown in blue and red, respectively). Top five (by protein count) significantly enriched Reactome terms shown. (**E**) Analyses of fatty acid (FA) composition reveal increased de novo lipogenesis (reflected by C16:0/C18:2 n ratio) and FA desaturation (reflected by C16:1 n-7/C16:0 ratio) in gWAT of *Nr1d1⁻/⁻*. n=6/group. Data presented as mean with individual data points (**A, E**). *p<0.05, **p<0.01, unpaired two-tailed t-test (**A, E**). Source data for panels C, D available in *Figure 1—source data 1*.

The online version of this article includes the following source data and figure supplement(s) for figure 1:

**Source data 1.** Source data (protein lists) for *Figure 1*, panels C, D.

**Figure supplement 1.** Rhythmic physiology and susceptibility to diet-induced obesity in *Nr1d1⁻/⁻* mice.

**Figure supplement 1—source data 1.** Raw uncropped and annotated blot images for *Figure 1—figure supplement 1*, panel F.

controls, although genotype differences in day/night activity and temperature levels suggest some dampening of rhythmicity in the *Nr1d1⁻/⁻* mice (*Figure 1—figure supplement 1A–E*). However, this is unlikely to account for the increased adiposity in these animals, and a previous study did not report significant genotype differences in these parameters (*Delezie et al., 2012*). This favours instead an altered energy partitioning within these mice, with a clear bias towards storing energy as lipid.

To explore further the lipid storage phenotype, we undertook proteomic analysis of gonadal white adipose tissue (gWAT) collected at ZT8 (zeitgeber time, 8 hr after lights on), the time of normal peak in NR1D1 expression in this tissue (*Figure 1—figure supplement 1F*). Isobaric tag (iTRAQ) labelled LC-MS/MS identified 2257 proteins, of which 182 demonstrated differential regulation (FDR<0.05) between WT and *Nr1d1⁻/⁻* gWAT samples (n=6 weight-matched, 13-week-old male mice/group) (*Figure 1C*). Differentially expressed (DE) proteins included influential metabolic enzymes, with up-regulation of metabolic processes detected on pathway enrichment analysis (*Figure 1D*). Importantly, and in line with the phenotype observed, increased NADPH regeneration (e.g. ME1, G6PDX), enhancement of glucose metabolism (also likely reflecting increased glyceroneogenesis, e.g. PFKL, ALDOA), and up-regulation of fatty acid synthesis (e.g. ACYL, FAS, ACACA) all support a shunt towards synthesis and storage of fatty acids and triglyceride in the KO mice. To

validate this putative increase in local lipid synthesis, we quantified fatty acid species in gWAT, and indeed, the ratio of palmitic to linoleic acid (C16:0/C18:2n6), a marker of de novo lipogenesis, was significantly elevated in *Nr1d1*[-/-] samples (*Figure 1E*, *Figure 1—figure supplement 1G*). Fatty acid profiling also revealed evidence of increased SCD1 activity (C16:1 n-7/C16:0). Enhanced fatty acid synthesis in gWAT of mice lacking NR1D1 may be in part driven by increased glucose availability and adipose tissue uptake as previously suggested (*Delezie et al., 2012*), although we do not observe elevated blood glucose levels in the *Nr1d1*[-/-] animals (*Figure 1—figure supplement 1H*). The propensity to lipid storage is further highlighted by the substantial obesity, compared to litter-mate controls, displayed by *Nr1d1*[-/-] mice when challenged with 10 weeks of HFD (*Figure 1—figure supplement 1I*; *Delezie et al., 2012*; *Hand et al., 2015*). Interestingly, we observed a strong positive correlation between body weight and daily intake of HFD in the *Nr1d1*[-/-] mice (*Figure 1—figure supplement 1J*), suggesting that HFD-induced hyperphagia exacerbates weight gain and obesity in the *Nr1d1*[-/-] mice.

## Limited impact of adipocyte-selective *Nr1d1* deletion under basal conditions

To define the role of NR1D1 specifically within WAT, we generated a new mouse line with loxP sites flanking *Nr1d1* exons 2–6 (*Nr1d1*[Flox2-6]), competent for Cre-mediated conditional deletion (*Hunter et al., 2020*). We crossed this mouse with the well-established *adiponectin* Cre-driver line (*Adipo*[Cre]; *Eguchi et al., 2011*; *Jeffery et al., 2014*) to delete *Nr1d1* selectively in adipocytes. This

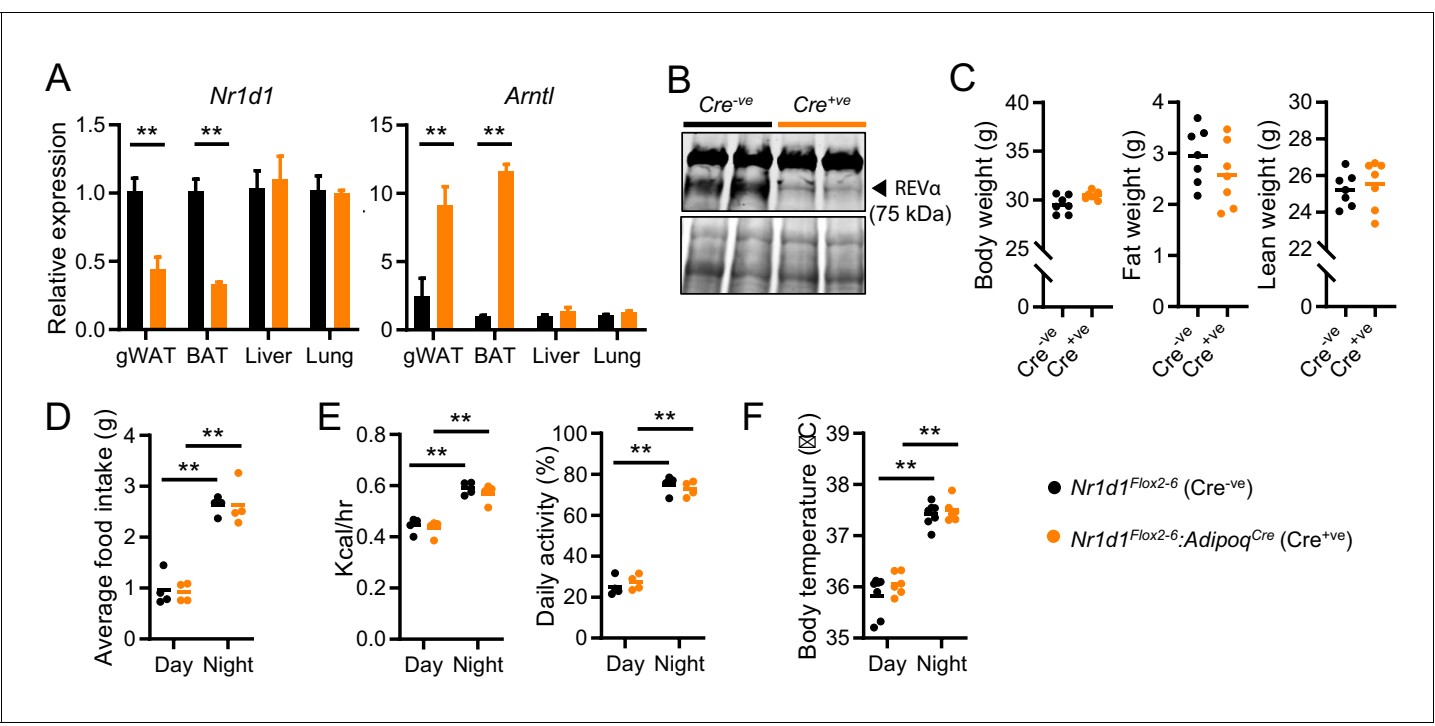

**Figure 2.** Impact of adipose *Nr1d1* deletion is limited under normal conditions. (**A**) *Nr1d1* and *Arntl* (*Bmal1*) gene expression in gonadal white adipose tissue (gWAT), brown adipose (BAT), liver and lung in *Nr1d1*[Flox2-6] (Cre[-ve]), and *Nr1d1*[Flox2-6]*:Adipoq*[Cre] (Cre[+ve]) mice (n=4–5/group). (**B**) NR1D1 protein expression (arrowhead) in Cre[-ve] and targeted Cre[+ve] mice. Lower blot shows Ponceau S protein staining. (**C**) Body weight, fat mass, and lean mass in 13-week-old Cre[-ve] and Cre[+ve] male mice (n=7/group). (**D–F**) Both *Nr1d1*[Flox2-6]*:Adipoq*[Cre] Cre[+ve] and Cre[-ve] mice demonstrate diurnal rhythms in behaviour and physiology, with no genotype differences observed in food intake (**D**), energy expenditure and daily activity (**E**) or body temperature (**F**) in 13-week-old males (n=4–7/group). Data presented as mean ± SEM (**A**) or as mean with individual data points (**C–F**). *p<0.05, **p<0.01, unpaired t-tests corrected for multiple comparisons (**A**), unpaired t-tests (**C**), two-way ANOVA with Tukey's multiple comparisons tests (**D–F**).

The online version of this article includes the following source data and figure supplement(s) for figure 2:

**Source data 1.** Raw uncropped and annotated blot images for *Figure 2*, panel B.

**Figure supplement 1.** Loss of *Nr1d1* expression in brown adipocytes does not alter body temperature.

**Figure supplement 1—source data 1.** Raw uncropped and annotated blot images for *Figure 2—figure supplement 1*, panel B.

new line results in loss of *Nr1d1* mRNA (*Figure 2A*) and protein (*Figure 2B*) expression in adipose tissue depots, as well as coordinate de-repression of *Arntl* (*Bmal1*), upon Cre-mediated recombination. In marked contrast to global *Nr1d1⁻/⁻* mice, adult *Nr1d1^Flox2-6^:Adipoq^Cre^* mice did not show an increase in adiposity when maintained on a standard chow diet (*Figure 2C*; n=7/group), with no differences in mean body weight, fat, and lean mass observed. In parallel with this, we saw no differences in daily patterns of food intake, energy expenditure, activity levels, or body temperature in matched *Nr1d1^Flox2-6^:Adipoq^Cre^* and control (*Nr1d1^Flox2-6^*) mice (*Figure 2D–F*). As brown adipose

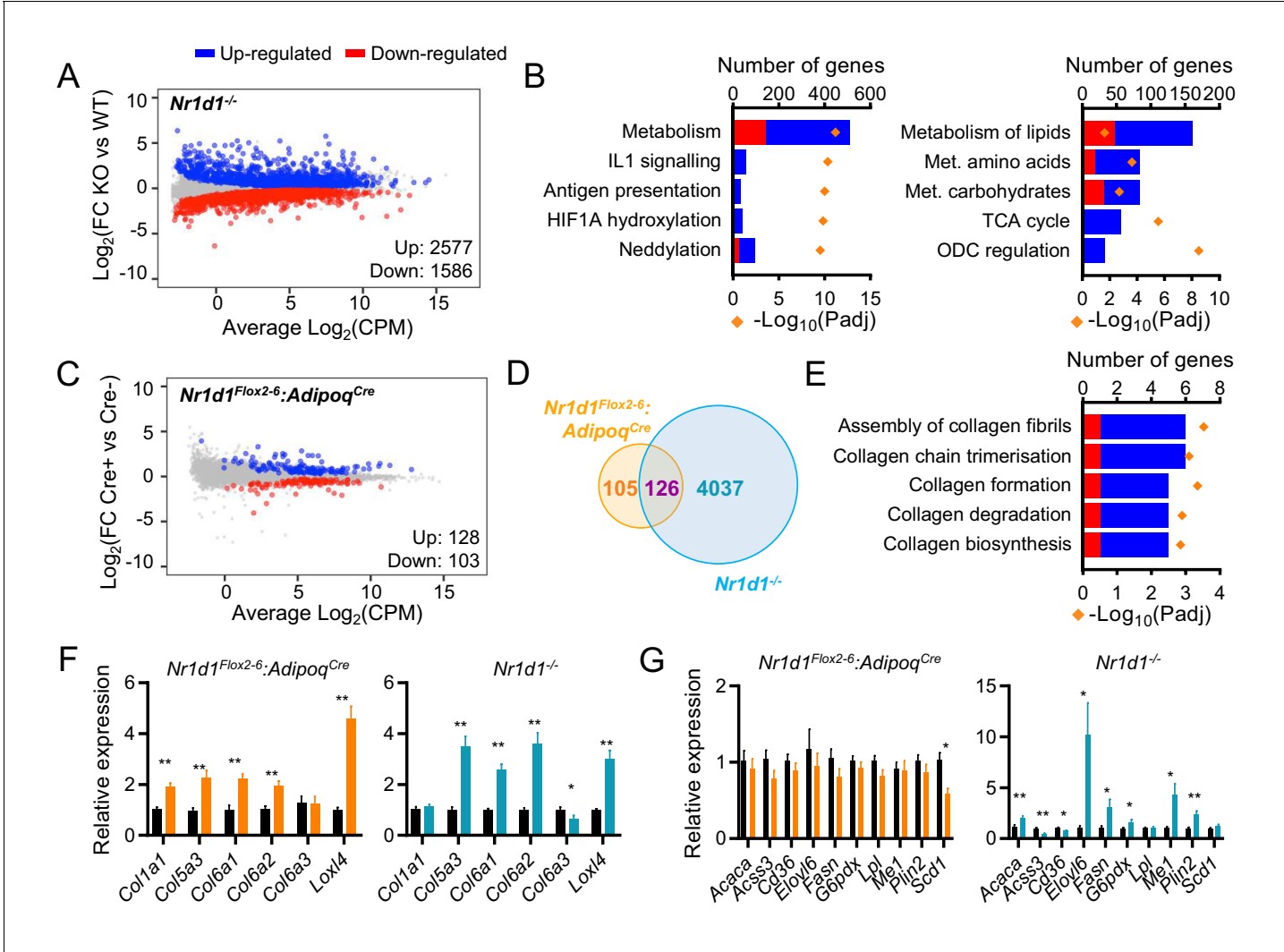

**Figure 3.** Global or adipose-specific *Nr1d1* deletion produces distinctive gene expression profiles. (A, B) *Nr1d1⁻/⁻* gonadal white adipose tissue (gWAT) demonstrates extensive remodelling of the transcriptome and up-regulation of metabolic pathways. Mean difference (MD) plot (A) showing significantly (FDR<0.05) up- (blue) and down- (red) regulated genes in gWAT of *Nr1d1⁻/⁻* mice compared to littermate controls (n=6/group). Pathway analysis (B) of significantly differentially expressed (DE) genes (FDR<0.05): top five (by gene count) significantly enriched Reactome terms shown (left panel), top five (by gene count) metabolic pathways shown (right panel). Up-regulated genes in blue, down-regulated in red. ODC = ornithine decarboxylase. (C) By contrast, RNA-seq demonstrates modest remodelling of the transcriptome in gWAT of *Nr1d1^Flox2-6^:Adipoq^Cre^* mice. MD plot, n=6/group. (D) Venn diagram showing overlap of DE genes in *Nr1d1^Flox2-6^:Adipoq^Cre^* and *Nr1d1⁻/⁻* gWAT. (E) Pathway analysis of 126 commonly DE genes. Top five (by gene count) significantly enriched Reactome terms shown. (F, G) Collagen genes are commonly up-regulated in both genotypes (F), whilst genes of lipid metabolism are not DE in *Nr1d1^Flox2-6^:Adipoq^Cre^* (G). gWAT qPCR, n=6–7/group. Data presented as mean ± SEM (F, G). *p<0.05, **p<0.01, unpaired t-tests corrected for multiple comparisons (F, G). Source data for panels A–E available in *Figure 3—source data 1*.

The online version of this article includes the following source data and figure supplement(s) for figure 3:

**Source data 1.** Source data (lists of differentially expressed genes, pathway lists) for *Figure 3*, panels A–E.
**Figure supplement 1.** Impact of *Nr1d1* and *Nr1d2* loss in vitro.

tissue (BAT) makes an important contribution to whole body energy metabolism, we studied thermo-regulation in $Nr1d1^{Flox2-6}$:$Adipoq^{Cre}$ mice in greater detail. It has previously been proposed that NR1D1 is key in conferring circadian control over thermogenesis, through its repression of uncoupling protein 1 (UCP1) expression (*Gerhart-Hines et al., 2013*). However, we saw no evidence of genotype differences in thermoregulation between $Nr1d1^{Flox2-6}$:$Adipoq^{Cre}$ and $Nr1d1^{Flox2-6}$ mice (*Figure 2—figure supplement 1A–E*). Despite increased BAT UCP1 expression in the Cre$^{+ve}$ mice, no differences in body temperature profiles were observed between Cre$^{-ve}$ and Cre$^{+ve}$ mice when housed under normal laboratory temperature (22°C) nor when placed under thermoneutral conditions (29°C) for >14 days. Moreover, Cre$^{-ve}$ and Cre$^{+ve}$ mice did not differ in their thermogenic response to an acute cold challenge (4°C for 6 hr) (*Figure 2—figure supplement 1F–H*). These data support our overall findings that adipocyte-selective deletion of $Nr1d1$ does not have a significant impact on metabolic phenotype under normal chow (NC) feeding conditions. Together, our data also suggest that the lack of adiposity phenotype in the Cre$^{+ve}$ mice is not driven by an increase in locomotor activity, energy expenditure, or thermogenesis.

## In normal WAT, NR1D1 regulated targets are limited to clock and collagen genes

To investigate adipocyte-specific $Nr1d1$ activity, we performed RNA-seq at ZT8 (n=6/group) in both $Nr1d1^{-/-}$ and $Nr1d1^{Flox2-6}$:$Adipoq^{Cre}$ mouse lines. Global $Nr1d1$ deletion had a large effect on the gWAT transcriptome, with 4163 genes showing significant differential expression (FDR¡0.05) between $Nr1d1^{-/-}$ mice and age- and weight-matched WT littermate controls (*Figure 3A*). Pathway enrichment analysis demonstrated that these changes are dominated by metabolic genes (*Figure 3B*), with lipid metabolism and the TCA cycle emerging as prominent processes (*Figure 3B*). Thus, the gWAT transcriptome in $Nr1d1^{-/-}$ mice is concordant with the phenotype, and the gWAT proteome, in demonstrating up-regulation of lipid accumulation and storage processes. By contrast, and consistent with the absence of an overt phenotype, only a small genotype effect on the transcriptome was observed when comparing gWAT RNA-seq from $Nr1d1^{Flox2-6}$:$Adipoq^{Cre}$ and $Nr1d1^{Flox2-6}$ littermates (*Figure 3C*; n=6/group). Here, 231 genes showed significant differential expression between genotypes, of which 126 were also differentially regulated in the WAT analysis of global $Nr1d1^{-/-}$ mice (*Figure 3D*). These 126 common genes included circadian clock components (*Arntl, Clock, Cry2, Nfil3*), whilst pathway analysis also revealed collagen formation/biosynthesis processes to be significantly enriched (*Figure 3E*). Regulation of the molecular clock is expected, but the discovery of collagen dynamics as a target of NR1D1 regulatory action in adipocytes has not been previously recognised. We validated consistent up-regulation of collagen and collagen-modifying genes in $Nr1d1^{-/-}$ and $Nr1d1^{Flox2-6}$:$Adipoq^{Cre}$ gWAT by qPCR (*Figure 3F*). It is notable that $Nr1d1^{Flox2-6}$:$Adipoq^{Cre}$ mouse gWAT exhibits neither enrichment of lipid metabolic pathways nor de-regulation of individual key lipogenic genes, previously identified as NR1D1 targets (*Figure 3E, G*; *Feng et al., 2011*; *Zhang et al., 2015*; *Zhang et al., 2016*). These findings suggest that lipogenic gene regulation may be a response to system-wide changes in energy metabolism in the $Nr1d1^{-/-}$ animals and challenge current understanding of NR1D1 action. These findings parallel our recent observations in hepatocyte-selective deletion of $Nr1d1$ (*Hunter et al., 2020*).

Work in liver has suggested that the NR1D1 paralogue, NR1D2 (REVERBβ), contributes to the suppression of lipogenesis, and that concurrent NR1D2 deletion amplifies the impact of NR1D1 loss (*Bugge et al., 2012*). We therefore performed double knockdown of $Nr1d1$ and $Nr1d2$ in differentiated 3T3-L1 cells (*Figure 3—figure supplement 1A*). Whilst double knockdown produced greater $Arntl$ de-repression than either $Nr1d1$ or $Nr1d2$ knockdown alone, it did not lead to de-repression of lipogenic genes previously considered NR1D1 targets (*Figure 3—figure supplement 1B*). We observed a similar pattern when comparing the liver transcriptome of liver-specific $Nr1d1/Nr1d2$ dual-deletion mice with that of global $Nr1d1^{-/-}$ KO; double knockdown produces a greater effect than loss of $Nr1d1$ alone, but does not lead to de-repression of metabolic genes (*Hunter et al., 2020*). Moreover, any potential compensation by NR1D2 does not prevent adipose lipid accumulation in the global $Nr1d1^{-/-}$ mice. Thus, although it cannot be ruled out, these findings suggest that compensation by NR1D2 does not underlie the mild phenotype observed in the $Nr1d1^{Flox2-6}$:$Adipoq^{Cre}$ mice.

Therefore, whilst global NR1D1 targeting produces an adiposity phenotype with up-regulation of WAT lipogenesis and lipid storage, this is not seen when NR1D1 is selectively targeted in adipose

alone. The distinction is not due to loss of *Nr1d1* expression in brown adipose, and is not due to compensatory NR1D2 action. Taken together, our data suggest that under a basal metabolic state, the adipose transcriptional targets under direct NR1D1 control are in fact limited to core clock function and collagen dynamics. NR1D1 is not a major repressor of lipid metabolism in this setting. This also suggests that the enhanced lipid accumulation phenotype of *Nr1d1*$^{-/-}$ adipose tissue is either independent from adipose NR1D1 entirely or that the action of NR1D1 in adipose is context-dependent.

## Diet-induced obesity reveals a broader WAT phenotype in tissue-specific NR1D1 deletion

Studies in liver tissue have demonstrated reprogramming of both nuclear receptor and circadian clock factor activity by metabolic challenge (*Eckel-Mahan et al., 2013*; *Goldstein et al., 2017*; *Guan et al., 2018*; *Quagliarini et al., 2019*). Both our data here, and previous reports (*Delezie et al., 2012*; *Feng et al., 2011*; *Hand et al., 2015*; *Le Martelot et al., 2009*; *Preitner et al., 2002*), highlight that the NC-fed *Nr1d1*$^{-/-}$ mouse is metabolically abnormal. The emergence of the collagen dynamics as a direct NR1D1 target and exaggerated diet-induced obesity evident in *Nr1d1*$^{-/-}$ mice supports a role for NR1D1 in regulating adipose tissue expansion under obesogenic conditions. To test this, *Nr1d1*$^{Flox2-6}$:*Adipoq*$^{Cre}$ and *Nr1d1*$^{Flox2-6}$ mice were provided with HFD for 16 weeks to drive obesity and WAT expansion. Indeed, compared to their controls, *Nr1d1*$^{Flox2-6}$:*Adipoq*$^{Cre}$ mice exhibited greater weight gain and adiposity in response to HFD feeding (*Figure 4A,B*). Of note, divergence between control and *Nr1d1*$^{Flox2-6}$:*Adipoq*$^{Cre}$ mice became clear only after long-term HFD feeding (beyond 13 weeks), a time at which body weight gain plateaus in control mice. This contrasts substantially with *Nr1d1*$^{-/-}$ mice, which show rapid and profound weight gain from the start of HFD feeding (*Hand et al., 2015*). The stark difference in progression and severity of diet-induced obesity is likely due (at least in part) to the HFD-induced hyperphagia, which is observed in *Nr1d1*$^{-/-}$ mice (WT food intake 2.92±0.10 g HFD/day/mouse; KO 3.74±0.21 g, p=0.0014, Student's t-test, n=21/genotype), but not in *Nr1d1*$^{Flox2-6}$:*Adipoq*$^{Cre}$ mice (Cre$^{-ve}$ 2.99±0.61 g HFD/day/mouse; Cre$^{+ve}$ 3.01±0.60 g, p>0.05, n=8/genotype). Nevertheless, both models highlight that loss of NR1D1 increases capacity for increased lipid storage and adipose tissue expansion under obesogenic conditions. Despite the enhanced diet-induced obesity, HFD-fed *Nr1d1*$^{Flox2-6}$:*Adipoq*$^{Cre}$ mice showed little evidence of typical obesity-related pathology. Histological assessment of gWAT after 16 weeks of HFD feeding revealed widespread adipose tissue fibrosis (Picrosirius Red staining of collagen deposition under normal and polarised light) and macrophage infiltration (F4/80 immunohistochemistry) in obese control mice, but these features were not seen in *Nr1d1*$^{Flox2-6}$:*Adipoq*$^{Cre}$ mice (*Figure 4C,D*). Furthermore, we saw evidence of preserved insulin sensitivity in the HFD-fed *Nr1d1*$^{Flox2-6}$:*Adipoq*$^{Cre}$ mice, with neither circulating glucose nor insulin being higher than *Nr1d1*$^{Flox2-6}$ littermate controls (*Figure 4E*), despite carrying significantly greater fat mass. Indeed, on insulin tolerance testing, HFD-fed *Nr1d1*$^{Flox2-6}$:*Adipoq*$^{Cre}$ mice demonstrated a significantly greater hypoglycaemic response than that observed in HFD-fed controls (*Figure 4F*). We saw no differences in adipocyte size between the two genotypes, indicating that our observations did not simply reflect greater adipocyte hypertrophy in the *Nr1d1*$^{Flox2-6}$:*Adipoq*$^{Cre}$ mice (*Figure 4—figure supplement 1A*). In line with a pronounced increase in fat mass, both gWAT and inguinal white adipose tissue depots (iWAT) were substantially larger in obese *Nr1d1*$^{Flox2-6}$:*Adipoq*$^{Cre}$ mice than in obese *Nr1d1*$^{Flox2-6}$ controls (*Figure 4—figure supplement 1B*).

Therefore, under long-term HFD-feeding conditions, adipose-targeted *Nr1d1* deletion results in continued adipose tissue expansion accompanied by a healthier metabolic phenotype with reduced adipose inflammation and fibrosis, and preserved systemic insulin sensitivity. Importantly, these findings also suggest that the regulatory influence of NR1D1 is context-dependent, with the metabolic impact of adipose-targeted Nr1d1 deletion revealed by transition to an obese state.

## NR1D1-dependent gene regulation is reprogrammed by obesity

We next performed RNA-seq on gWAT collected at ZT8 from *Nr1d1*$^{Flox2-6}$:*Adipoq*$^{Cre}$ and *Nr1d1*$^{Flox2-6}$ littermate controls fed either NC or HFD for 16 weeks (NC, n=4/group; HFD, n=6/group). As expected, HFD feeding had a substantial impact on the gWAT transcriptome in both Cre$^{-ve}$ and Cre$^{+ve}$ animals (i.e. NC vs. HFD comparison within each genotype; *Figure 5A*). Under NC

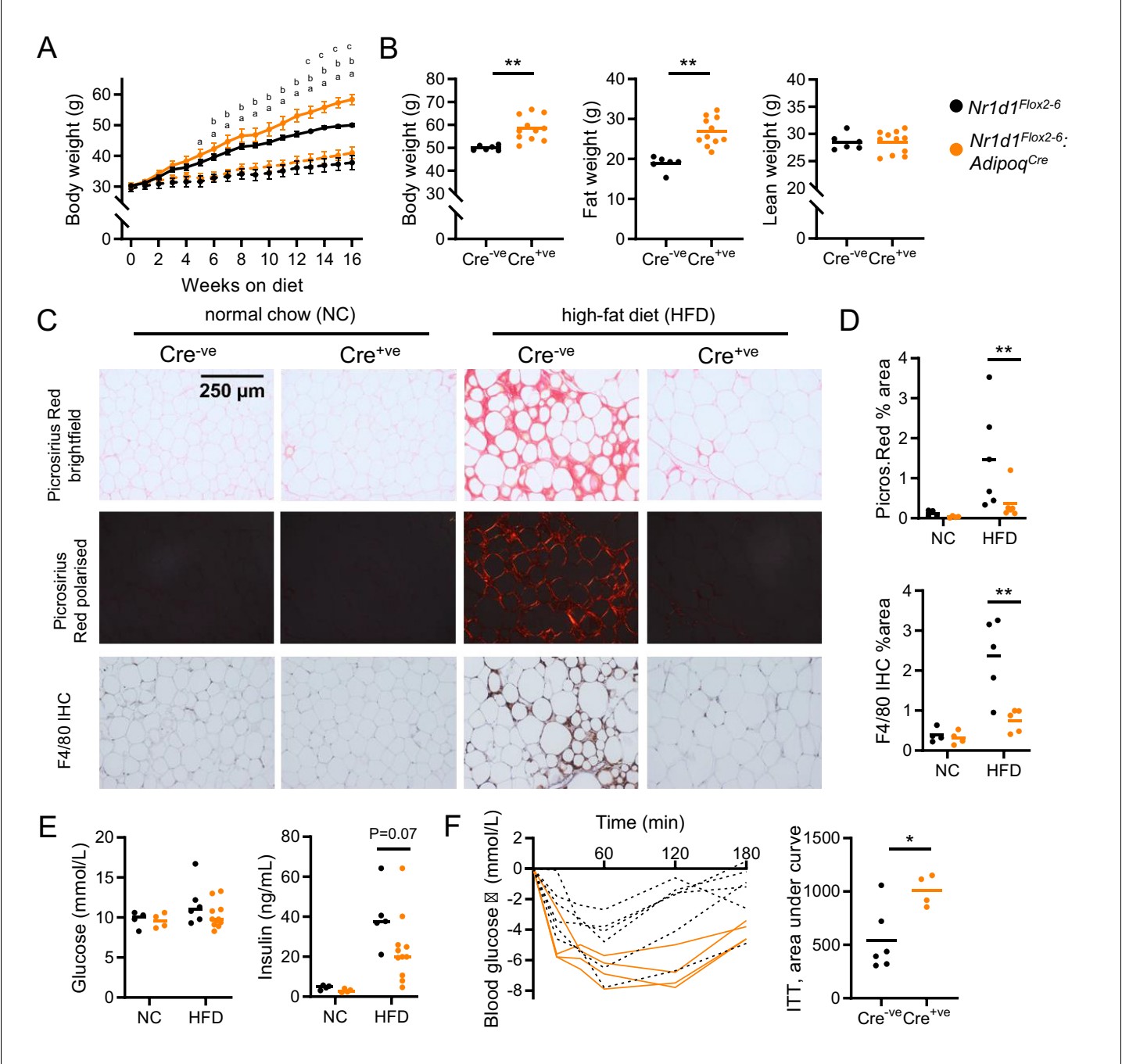

**Figure 4.** Diet-induced obesity unmasks a role for NR1D1 in the regulation of adipose expansion. (**A, B**) High-fat diet (HFD) leads to exaggerated adiposity in *Nr1d1*$^{Flox2-6}$:*Adipoq*$^{Cre}$ mice. Body weight track of Cre$^{-ve}$ and Cre$^{+ve}$ male mice on HFD (solid line) or normal chow (NC) (dashed line) (**A**) ($^a$p<0.05: Cre$^{+ve}$ NC vs. HFD; $^b$p<0.05, Cre$^{-ve}$ NC vs. HFD; $^c$p<0.05, Cre$^{+ve}$ HFD vs. Cre$^{-ve}$ HFD); total body, fat, and lean weight after 16 weeks in the high-fat diet group (**B**). (**C, D**) On histological examination of gonadal white adipose tissue (gWAT), HFD-fed Cre$^{+ve}$ mice display less fibrosis and inflammation than Cre$^{-ve}$ littermates. Representative Picrosirius Red and F4/80 immunohistochemistry images (20× magnification) (**C**), quantification of staining across groups, each data point represents the mean value for each individual animal (**D**). (**E, F**) Despite increased adiposity, HFD-fed Cre$^{+ve}$ mice display greater insulin sensitivity than Cre$^{-ve}$ controls. Terminal blood glucose and insulin levels (animals culled 2 hr after food withdrawal) in NC and HFD-fed in *Nr1d1*$^{Flox2-6}$:*Adipoq*$^{Cre}$ Cre$^{-ve}$ (black) and Cre$^{-ve}$ (orange) mice (**E**). Blood glucose values for individual animals and area under curve (change from baseline) for 16-week HFD-fed *Nr1d1*$^{Flox2-6}$:*Adipoq*$^{Cre}$ Cre$^{-ve}$ and Cre$^{-ve}$ mice undergoing insulin tolerance testing (ITT) (**F**). Data presented as mean ± SEM (**A**) or as individual data points with mean (**B, D, E, F**). *p<0.05, **p<0.01, two-way repeated-measures ANOVA with Tukey's multiple comparisons tests (**A**), two-way ANOVA with Sidak's multiple comparisons tests (**D, E**), unpaired two-tailed t-test (**B, F**). n=4–11/group for all panels. Picrosirius Red images for each animal available in *Figure 4—source data 1*.

*Figure 4 continued on next page*

*Figure 4 continued*

The online version of this article includes the following source data and figure supplement(s) for figure 4:

**Source data 1.** Source data (Picrosirius Red images, one per animal) for *Figure 4*, panel C.
**Figure supplement 1.** Adipose characteristics in *Nr1d1*$^{Flox2-6}$ and *Nr1d1*$^{Flox2-6}$:*Adipoq*$^{Cre}$ mice.

feeding conditions, we again observed only a small genotype effect on the transcriptome, and as before, DE genes included core clock genes (*Arntl, Nfil3, Npas2, Clock*) and those of collagen synthesis pathways (*Figure 5B*). However, obesity revealed a substantial genotype effect, with 3061 genes DE (1706 up, 1355 down) in HFD-fed *Nr1d1*$^{Flox2-6}$:*Adipoq*$^{Cre}$ mice vs. HFD-fed *Nr1d1*$^{Flox2-6}$ controls (*Figure 5B*), and 1704 genes showing a significant (α<0.05) diet-genotype interaction (stageR-specific interaction analysis; *Van den Berge et al., 2017*). Of these 1704 genes, those up-regulated in obese *Nr1d1*-deficient adipose were strongly enriched for metabolic pathways, whilst down-regulated genes showed weak enrichment of ECM organisation processes (*Figure 5C*). To examine how loss of *Nr1d1* alters adipose tissue response to diet-induced obesity, we compared directly those processes which showed significant obesity-related dysregulation in control mice (*Figure 5D*). While HFD feeding caused a profound down-regulation (vs. NC conditions) of metabolic pathways in the WAT of control mice, this was not observed in *Nr1d1*$^{Flox2-6}$:*Adipoq*$^{Cre}$ mice. HFD feeding led to an up-regulation of immune pathways in both genotypes (*Figure 5D*); however, as highlighted by Ingenuity Pathway Analysis (IPA) of differentially regulated genes in gWAT, inflammation and immune-related processes were widely and markedly attenuated in the HFD-fed Cre$^{+ve}$ mice when compared to HFD-fed controls (*Figure 5—source data 1*). Thus, transcriptomic profiling correlates with phenotype in suggesting that WAT function and metabolic activity is protected from obesity-related dysfunction in the *Nr1d1*$^{Flox2-6}$:*Adipoq*$^{Cre}$ mice, and that the impact of adipose *Nr1d1* deletion is dependent on system-wide metabolic state.

## Integration of differential gene expression with the WAT cistrome reveals state-dependent regulation of metabolic targets by NR1D1

To understand this protective effect of *Nr1d1* deletion, we profiled the gWAT NR1D1 cistrome using HaloTag-based technology. This permits antibody-independent capture of the NR1D1 cistrome, offering superior sensitivity and specificity to antibody-based approaches (*Hunter et al., 2020*). HaloChIP-seq was performed in gWAT tissue collected from NC-fed *HaloNr1d1* mice at ZT8 and at ZT20, and from WT mice at ZT8. ZT8 and ZT20 are the expected peak and nadir of NR1D1 recruitment to the genome, respectively (*Cho et al., 2012*; *Feng et al., 2011*). We identified 4474 ZT8 HaloNR1D1 peaks (MACS2, q<0.01) commonly called against both Halo ZT20 and WT ZT8 libraries (*Figure 6A*). As anticipated, we saw clear evidence of NR1D1 binding in proximity to core clock genes (*Figure 6—figure supplement 1A*), with this signal not seen in controls. Motif analysis of this high-confidence peak set detected strong enrichment of RORE and RevDR2 motifs (*Figure 6—figure supplement 1B*), again supporting the specificity of our cistrome.

The NR1D1 cistrome spanning >4000 binding sites contrasts with the small number of DE genes observed between *Nr1d1*$^{Flox2-6}$:*Adipoq*$^{Cre}$ and control gWAT under NC conditions (231 genes in the first RNA-seq experiment; *Figure 3C*, and 138 genes in the second; *Figure 5B*), but is compatible with the larger number of DE genes revealed under HFD conditions (*Figure 3A*, *Figure 5A*). We next used this NR1D1 cistrome to define the relationship between direct chromatin binding and altered gene expression under different genetic and/or dietary challenge. For this, we employed a custom Python script that calculates the enrichment of DE gene sets in spatial relation to identified transcription factor binding sites, over all genes in the genome (*Briggs et al., 2021b*; *Hunter et al., 2020*; *Yang et al., 2019*). We identified putative NR1D1 target genes by comparing gene sets which changed in both *Nr1d1*$^{Flox2-6}$:*Adipoq*$^{Cre}$ gWAT (relative to control mice, under both NC and HFD conditions) and in *Nr1d1*$^{-/-}$ gWAT (relative to WT littermates). Under NC conditions, only small sets of genes were up- or down-regulated in both *Nr1d1*$^{Flox2-6}$:*Adipoq*$^{Cre}$ and *Nr1d1*$^{-/-}$ tissues (vs. their respective controls) (*Figure 6B*). Extending out from NR1D1 ChIP-seq peaks by increasing distances (λ), we found transcription start sites (TSSs) of genes up-regulated in *Nr1d1*$^{Flox2-6}$:*Adipoq*$^{Cre}$ and *Nr1d1*$^{-/-}$ gWAT to be significantly enriched (above all genes in the genome) at distances up to 100 kbp (*Figure 6B*). This is consistent with repression mediated directly by DNA-bound NR1D1 and

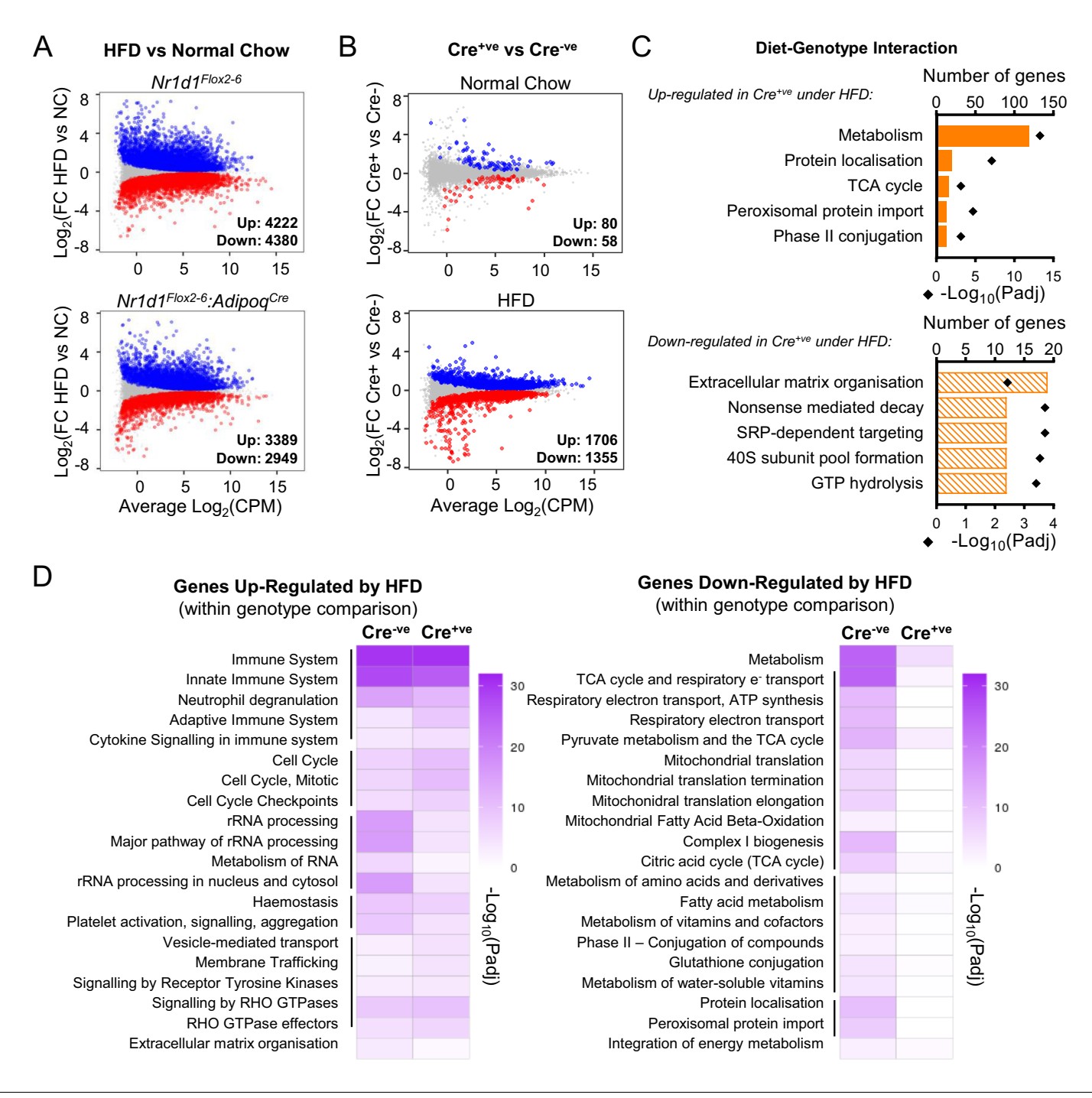

**Figure 5.** Under conditions of obesity, a broader programme of NR1D1 repression is seen. (A) High-fat diet (HFD) dramatically remodels the white adipose tissue (WAT) transcriptome. RNA-seq (n=4–6/group) was performed in gonadal WAT (gWAT) from Cre[-ve] and Cre[+ve] male mice fed normal chow (NC) or HFD for 16 weeks. Mean difference (MD) plots show genes significantly (FDR<0.05) up-regulated (blue) or down-regulated (red) by HFD in each genotype. (B) With HFD, the NR1D1-responsive gWAT transcriptome broadens. MD plots show effect of genotype in NC (top panel) and HFD (lower panel) feeding conditions. Genes where stageR detects a significant (α=0.05) genotype-diet interaction highlighted in orange. (C) Reactome pathway analysis of genes up- or down-regulated in Cre[+ve] gWAT under HFD conditions, where this diet-genotype interaction is also detected. Top five (by gene count) significantly enriched terms shown. (D) Adipose-targeted deletion of *Nr1d1* attenuates the normal HFD-induced down-regulation of metabolic pathways. Heatmaps show enrichment (-log10(padj)) of Reactome pathways in genes up-regulated (left) or down-regulated (right) by HFD feeding in Cre[-ve] and Cre[+ve] gWAT. Top 20 (by gene count in Cre[-ve] group) significantly enriched terms shown. Lines indicate related terms. Source data for panels A–D available in **Figure 5—source data 1**.

*Figure 5 continued on next page*

*Figure 5 continued*

The online version of this article includes the following source data for figure 5:

**Source data 1.** Source data (lists of differentially expressed genes, pathway lists) for *Figure 5*, panels A–D, plus IPA.

strongly suggests that this gene cluster (clock and collagen genes) represents direct targets of NR1D1 repression in WAT under NC conditions. In contrast, no enrichment of genes with decreased expression in *Nr1d1^Flox2-6^:Adipoq^Cre^* or *Nr1d1^-/-^* WAT was evident at any distance from NR1D1 peaks (*Figure 6B*). Thus, NR1D1 activation of transcription involves a different mechanism of regulation, likely involving secondary or indirect mechanisms (e.g. de-repression of another repressor), as previously proposed to explain NR1D1 transactivation (*Le Martelot et al., 2009*).

HFD -feeding of *Nr1d1^Flox2-6^:Adipoq^Cre^* mice greatly increased the overlap of DE genes with those DE in *Nr1d1^-/-^* WAT (*Figure 6C*). Critically, we observed a highly significant proximity enrichment of these commonly up-regulated genes (863) to sites of NR1D1 chromatin binding (*Figure 6C, D*), but again, saw no enrichment of the commonly down-regulated genes (*Figure 6C*). Thus, this integration of transcriptome and cistrome profiling suggests that these commonly up-regulated genes represent direct NR1D1 targets unmasked by the abnormal environment of obese adipose, and that NR1D1's exertion of direct repressive control is dependent on metabolic state.

Consistent with the healthier metabolic phenotype observed in obese *Nr1d1^Flox2-6^:Adipoq^Cre^* mice, we found that the large majority of the 863 NR1D1 gene targets unmasked by HFD feeding are normally repressed in obesity (*Figure 6E*), with 551 (63.8%) showing a significant down-regulation in obese control (Cre^-ve^) animals (chronic HFD compared to NC-fed state). Furthermore, 495 of these genes were found to lie within 100 kbp of a NR1D1 binding site (*Figure 6E*). These genes include important regulators of lipid and mitochondrial metabolism (*Figure 6F*) – including *Fasn*, *Scd1*, *Acsl1*, *Cs* – and FGF-21 co-receptor *Klb*, a previously identified NR1D1 target gene (*Jager et al., 2016*). Importantly, obesity-dependent de-repression of these genes was also observed in isolated mature adipocytes (MA) collected from NC or HFD-fed mice (*Figure 6—figure supplement 1C,D*). This provides further evidence that they are regulated by adipocyte NR1D1, and that this regulation is direct.

Considered together, these findings suggest that the healthy adiposity phenotype in HFD-fed *Nr1d1^Flox2-6^:Adipoq^Cre^* mice results from de-repression of NR1D1-controlled adipocyte pathways which allow continued and efficient lipid synthesis and storage, thus permitting greater expansion of the adipose bed, and attenuation of obesity-related dysfunction.

## Discussion

We set out to define the role of NR1D1 in the regulation of WAT metabolism and subsequently reveal a new understanding of NR1D1 function. Together, our data show NR1D1 to be a state-dependent regulator of WAT metabolism, with its widespread repressive action only unmasked by diet-induced obesity. Surprisingly, *Nr1d1* expression in WAT appears to limit the energy-buffering function of the tissue. This finding parallels our recent work in the liver (*Hunter et al., 2020*). Hepatic-selective loss of NR1D1 carries no metabolic consequence, with NR1D1-dependent control over hepatic energy metabolism revealed only upon altered feeding conditions. Contrary to current understanding, our findings therefore suggest that NR1D1 (and potentially other components of the peripheral clock) does not impose rhythmic repression of metabolic circuits under basal conditions but rather determines tissue responses to altered metabolic state. As reported previously by us and others (*Delezie et al., 2012*; *Hand et al., 2015*), global deletion of *Nr1d1* leads to an increase in lipogenesis, adipose tissue expansion, and an exaggerated response to diet-induced obesity. How the loss of NR1D1 specifically in WAT contributes to this phenotype has not previously been addressed. Here, we use proteomic, transcriptomic, and lipid profiling studies to show a clear bias towards fatty acid synthesis and triglyceride storage within *Nr1d1^-/-^* WAT. Adipocyte-targeted deletion of *Nr1d1* reveals only a modest phenotype, and a relatively selective set of gene targets, limited to clock processes and collagen dynamics. These genes are concurrently de-regulated in *Nr1d1^-/-^* adipose, and are found in proximity to NR1D1 binding sites, strongly implicating them as direct targets of NR1D1 repressive activity.

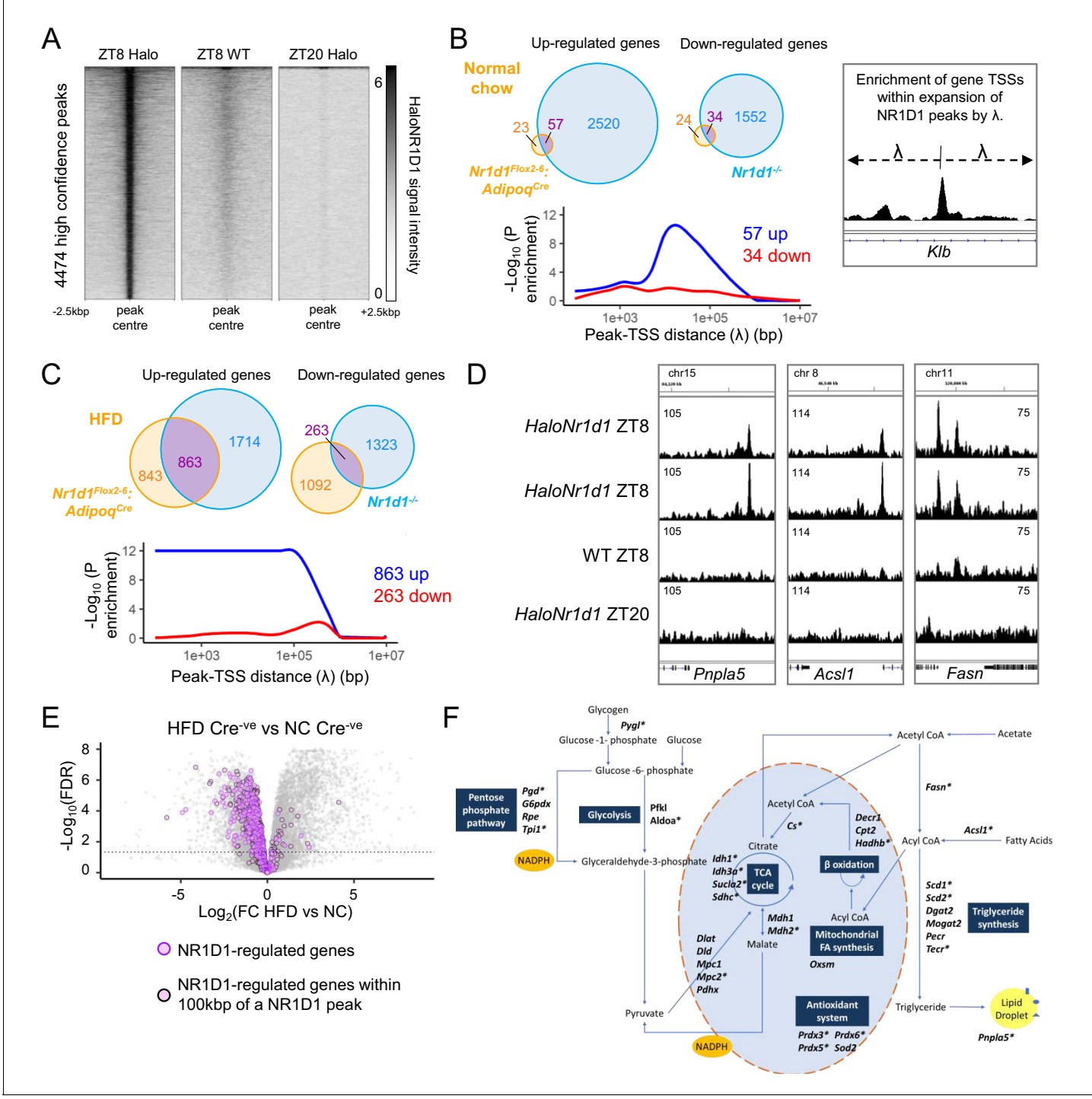

**Figure 6.** NR1D1 binding sites associate with genes of lipid and mitochondrial metabolism normally repressed in obese adipose. (**A**) Calling peaks against both wild-type (WT) ZT8 (zeitgeber time, 8 hr after lights on) and *HaloNr1d1* ZT20 samples detects 4474 high-confidence regions of HaloNR1D1 binding in gonadal white adipose tissue (gWAT). (**B**, **C**) Genes commonly up-regulated in normal chow (NC)-fed *Nr1d1^Flox2-6^:Adipoq^Cre^* and *Nr1d1^-/-^* gWAT (compared to littermate controls) are significantly enriched in proximity to HaloNR1D1 ChIP-seq peaks (**B**), as too are genes commonly up-regulated in high-fat diet (HFD)-fed *Nr1d1^Flox2-6^:Adipoq^Cre^* and *Nr1d1^-/-^* gWAT (**C**). Genes commonly down-regulated are not. Venn diagrams show intersection of up- and down-regulated genes in the two models of *Nr1d1* deletion. Plots shows enrichment (over all genes in the genome) of gene clusters of interest at increasing distances (λ) from HaloNR1D1 peaks. Up-regulated genes in blue, down-regulated in red. (**D**) Integrative Genomics Viewer (IGV) visualisations show HaloNR1D1 peaks in proximity to exemplar genes only up-regulated by *Nr1d1* deletion in obese adipose. Uniform y-axes within each panel. (**E**) NR1D1 targets are also down-regulated in obesity. Volcano plot highlighting effect of HFD (in intact (Cre^-ve^) animals) of the 863 NR1D1 target genes from (**C**). NR1D1 target genes shown in purple; 495 NR1D1 target genes also within 100 kbp of a HaloNR1D1 peak outlined in

*Figure 6 continued on next page*

*Figure 6 continued*

black. (F) Metabolic map illustrating NR1D1 targets. Genes with a transcription start site (TSS) within 100 kbp of a HaloNR1D1 ChIP-seq peak are starred*. Source data for panels A–C available in *Figure 6—source data 1*.

The online version of this article includes the following source data and figure supplement(s) for figure 6:

**Source data 1.** Source data (peak list, gene lists underlying Venn diagrams) for *Figure 6*, panels A–C.

**Figure supplement 1.** NR1D1 binding sites and gene expression changes in *Nr1d1^{Flox2-6}:Adipoq^{Cre}* mature adipocytes.

The reduced inflammation seen in obese *Nr1d1^{Flox2-6}:Adipoq^{Cre}* mice is likely multifactorial, but may be secondary to a reduction in the pro-inflammatory free fatty acid pool, resulting from de-repression of lipogenic and mitochondrial metabolism pathways, or the absence of signals from dead/dying adipocytes. It is of note that improved metabolic flexibility is beneficial in other mouse models of metabolic disease (*Jonker et al., 2012*; *Kim et al., 2007*; *Virtue et al., 2018*) and in human obesity (*Aucouturier et al., 2011*; *Begaye et al., 2020*). Whilst the clock has been linked to ECM remodelling in other tissues (*Chang et al., 2020*; *Dudek et al., 2016*; *Sherratt et al., 2019*), where it is thought to coordinate ECM dynamics, collagen turnover, and secretory processes (*Chang et al., 2020*), we now identify ECM as a direct target of adipose NR1D1 action; altered regulation of WAT collagen production and modification likely contributing to the rapid and continued adipose tissue expansion observed in *Nr1d1*-targeted adipose. The diminished adipose fibrosis seen in *Nr1d1^{Flox2-6}:Adipoq^{Cre}* mice may well reflect alteration in multiple processes, some of which may be under direct/indirect NR1D1 control. Adipose-specific deletion of *Nr1d1* thus provides a unique model to explore the complex ECM responses which accompany obesity-related tissue hypertrophy and development of fibrosis.

A role for the clock in the regulation of WAT function has been reported in the literature (*Barnea et al., 2015*; *Paschos et al., 2012*; *Shostak et al., 2013*), perhaps implying that it is the rhythmicity conferred by the clock which is important for WAT metabolism. However, despite robust rhythms of clock genes persisting, rhythmic gene expression in gWAT is largely attenuated following genetic disruption of SCN function (*Kolbe et al., 2016*). This supports the alternative notion that an intact local clock is not the primary driver of rhythmic peripheral tissue metabolism. Indeed, metabolic processes, including lipid biosynthesis, were highly enriched in the cohort of SCN-dependent rhythmic genes from this study (*Kolbe et al., 2016*), implying that feeding behaviour and WAT responses to energy flux are more important than locally generated rhythmicity for adipose function.

The modest impact of adipose-selective *Nr1d1* deletion is both at odds with the large effect of global *Nr1d1* deletion, and with the extensive WAT cistrome we have identified for NR1D1. This suggests that the tissue-specific actions of NR1D1 are necessary, but not sufficient, and require additional regulation from the metabolic state. Although not explored to the same extent, a lipogenic phenotype of liver-specific RORα/γ deletion has previously been shown to be unmasked by HFD feeding (*Zhang et al., 2017*). By driving adipose tissue hypertrophy through HFD feeding of the *Nr1d1^{Flox2-6}:Adipoq^{Cre}*, we observed a stark difference in the adipose phenotype of targeted mice and littermate controls. WAT tissue lacking *Nr1d1* showed significantly increased tissue expansion, but little evidence of normal obesity-related pathology (tissue fibrosis and immune cell infiltration/inflammation). Genes controlling mitochondrial activity, lipogenesis, and lipid storage were relatively spared from the obesity-related down-regulation observed in control mouse tissue, and were associated with the WAT NR1D1 cistrome. Thus, in response to HFD feeding, NR1D1 acts to repress metabolic activity in the adipocyte and limit tissue expansion (albeit at the eventual cost of tissue dysfunction, inflammation, and development of adipose fibrosis).

The broadening of NR1D1's regulatory influence in response to obesity likely reflects a change to the chromatin environment in which NR1D1 operates (*Figure 6—figure supplement 1E*). The majority of emergent NR1D1 target genes are repressed in obese adipose when *Nr1d1* expression is intact (*Figure 6E*). As these genes are not de-repressed by *Nr1d1* loss in normal adipose, NR1D1 activity must be redundant or ineffective in a 'basal' metabolic state. Subsequent emergence of NR1D1's transcriptional control may reflect alterations in chromatin accessibility or organisation, and/or the presence of transcriptional repressors and accessory factors required for full activity. As NR1D1 is itself proposed to regulate enhancer-promoter loop formation (*Kim et al., 2018*), modulation of *Nr1d1* expression would be a further important variable here. Such reshaping of the

regulatory landscape likely occurs across tissues, and may explain why, with metabolic challenge, emergent circadian rhythmicity is observed in gene expression (*Eckel-Mahan et al., 2013*; *Kinouchi et al., 2018*; *Tognini et al., 2017*), and in circulating and tissue metabolites (*Dyar et al., 2018*; *Eckel-Mahan et al., 2013*).

Here, we now uncover a role for NR1D1 in limiting the energy-buffering role of WAT, a discovery which may present therapeutic opportunity as we cope with an epidemic of human obesity. Despite recent findings which have cast doubt on the utility of some of the small molecule NR1D1 ligands (*Dierickx et al., 2019*), antagonising WAT NR1D1 now emerges as a potential target in metabolic disease. Finally, our study suggests that a functioning circadian clock may be beneficial in coping with acute mistimed metabolic cues but, that under chronic energy excess, may contribute to metabolic dysfunction and obesity-related pathology.

## Materials and methods

### Animal experiments

All experiments described here were conducted in accordance with local requirements and licenced under the UK Animals (Scientific Procedures) Act 1986, project licence number 70/8558 (DAB). Procedures were approved by the University of Manchester Animal Welfare and Ethical Review Body (AWERB). Unless otherwise specified, all animals had ad libitum access to standard laboratory chow and water, and were group-housed on 12 hr:12 hr light:dark cycles and ambient temperature of 22±1.5°C. Unless otherwise stated, male mice (*Mus musculus*) were used for all experimental procedures. All proteomics studies were carried out on 13-week-old weight-matched males. RNA-seq studies for *Figure 3* were carried out on 12- to 14-week-old weight-matched males; the RNA-seq study for *Figure 5* was carried out on 28-week-old males (following 16 weeks of HFD or NC feeding). HaloChIP-seq was carried out on males aged 12–21 weeks.

For group allocation, we employed a timed breeding approach, so that cohorts of mice were produced within a narrow time window. All studies compared littermate control and transgenic mice, which inherently confers group allocation and randomisation in cage housing (and thus diet regime). The n numbers, and overall experimental design, were determined on the basis of extensive experience with the models, and power analyses incorporating previous results (based on achieving 80% power with 5% type I error). Blinding was facilitated by animal numbering, numbered coding of samples at collection, and use of automated analyses where possible.

### Nr1d1$^{-/-}$

Nr1d1$^{-/-}$ mice were originally generated by Ueli Schibler (University of Geneva) (*Preitner et al., 2002*). These mice were created by replacing exons 2–5 of the *Nr1d1* gene by an in-frame LacZ allele. Mice were then imported to the University of Manchester and backcrossed to C57BL/6J mice.

### Nr1d1$^{Flox2-6}$

A CRISPR-Cas9 approach was used to generate a conditional knock allele for *Nr1d1* -, as described (*Hunter et al., 2020*). LoxP sites were integrated, in a two-step process, at intron 2 and intron 6, taking care to avoid any previously described transcriptional regulatory sites (*Yamamoto et al., 2004*). A founder animal with successful integration of both the 5' and 3' loxP sites, transmitting to the germline, was identified and bred forward to establish a colony.

### Nr1d1$^{Flox2-6}$:Adipoq$^{Cre}$

Adiponectin-driven Cre-recombinase mice (*Eguchi et al., 2011*; *Jeffery et al., 2014*) were purchased from the Jackson Laboratory and subsequently bred against the Nr1d1$^{Flox2-6}$ at the University of Manchester.

### HaloNr1d1

HaloNr1d1 mice were generated by the University of Manchester Genome Editing Unit, as described (*Hunter et al., 2020*).

## In vivo phenotyping

Body composition of mice was analysed prior to cull by quantitative magnetic resonance (EchoMRI 900). Energy expenditure was measured via indirect calorimetry using CLAMS (Columbus Instruments) for 10- to 12-week-old male mice. Mice were allowed to acclimatise to the cages for 2 days, prior to an average of 5 days of recordings being collected. Recording of body temperature and activity was carried out via surgically implanted radiotelemetry devices (TA-F10, Data Sciences International). Data is shown as a representative day average of single-housed age-matched males. For the diet challenge, male mice were fed HFD (60% energy from fat; DIO Rodent Purified Diet, IPS Ltd) for a period of 10–16 weeks from 12 weeks of age. Blood glucose was measured from tail blood using the Aviva Accuchek meter (Roche). For the insulin tolerance test, mice were fasted from ZT0, then injected with 0.75 IU/kg human recombinant insulin (I2643, Sigma-Aldrich) at ZT6 (time '0 min').

## Insulin ELISA

Insulin concentrations were measured by ELISA (EZRMI-13K Rat/Mouse insulin ELISA, Merck Millipore) according to the manufacturer's instructions. Samples were diluted in matrix solution to fall within the range of the assay. Internal controls supplied with the kit were run alongside the samples and were in the expected range.

## Histology

gWAT was collected and immediately fixed in 4% paraformaldehyde for 24 hr, transferred into 70% ethanol, and processed using a Leica ASP300 S tissue processor; 5 µm sections underwent H and E staining (Alcoholic Eosin Y solution [HT110116] and Harris Haematoxylin solution [HHS16], Sigma-Aldrich), Picrosirius Red staining (see below), or F4/80 immunohistochemistry (see below). Images were collected on an Olympus BX63 upright microscope using 10×/0.40 UPlan SAPo and 20×/0.75 UApo/340 objectives. Percentage area stained was quantified using ImageJ (version 1.52a) as detailed in the online ImageJ documentation, with 5–12 images quantified per animal. Adipocyte area was quantified using the Adiposoft ImageJ plug-in (version 1.16).

For Picrosirius Red staining, sections were dewaxed and rehydrated using the Leica ST5010 Autostainer XL. Sections were washed in distilled water and then transferred to Picrosirius Red (Direct Red 80, Sigma-Aldrich) (without the counterstain) for 1 hr. Sections were then washed briefly in 1% acetic acid. Sections were then dehydrated, cleared, and mounted using the Leica ST5010 Autostainer XL.

For F4/80 immunohistochemistry, sections were dewaxed and rehydrated prior to enzymatic antigen retrieval (trypsin from porcine pancreas [T7168, Sigma]). Sections were treated with 3% hydrogen peroxide to block endogenous peroxidase activity followed by further blocking with 5% goat serum. Rat mAb to F4/80 (1:500) (ab6640, Abcam) was added and sections were incubated overnight at 4°C. Sections were washed before addition of the biotinylated anti-rat IgG (BA-9400, H and L) secondary antibody (1:1500) for 1 hr. Sections were developed using VECTAstain Elite ABC kit peroxidase, followed by DAB Peroxidase substrate (Vector Labs) and counterstained with haematoxylin. Slides were then dehydrated, cleared, and mounted.

## Lipid extraction and gas chromatography

Total lipid was extracted from tissue lysates using chloroform-methanol (2:1; v/v) according to the Folch method (*Folch et al., 1957*). An internal standard (tripentadecanoin glycerol [15:0]) of known concentration was added to samples for quantification of total triacylglyceride. Lipid fractions were separated by solid-phase extraction and fatty acid methyl esters (FAMEs) were prepared as previously described (*Heath et al., 2003*). Separation and detection of total triglyceride FAMEs was achieved using a 6890N Network GC System (Agilent Technologies, Santa Clara, CA) with flame ionisation detection. FAMEs were identified by their retention times compared to a standard containing 31 known fatty acids and quantified in micromolar from the peak area based on their molecular weight. The micromolar quantities were then totalled and each fatty acid was expressed as a percentage of this value (molar percentage, mol%).

## Proteomics

Mice were culled by cervical dislocation and the gWAT was immediately removed and washed twice in ice-cold PBS and then once in ice-cold 0.25 M sucrose, prior to samples being snap-frozen in liquid nitrogen and stored at −80℃. To extract the protein, the samples were briefly defrosted on ice and then cut into 50 mg pieces and washed again in ice-cold PBS. The sample was then lysed in 200 μl of 1 M triethylammonium bicarbonate buffer (Sigma) with 0.1% (w/v) sodium dodecyl sulphate with a Tissue Ruptor (Qiagen). Samples were centrifuged for 5 min, full speed, at 4℃ and the supernatant collected into a clean tube. A Bio-Rad Protein Assay (Bio-Rad) was used to quantify the protein and Coomassie protein stain (InstantBlue Protein Stain Instant Blue, Expedeon) to check the quality of extraction. Full methods of subsequent iTRAQ proteomic analysis including bioinformatic analysis has been published previously (*Kassab et al., 2019*; *Xu et al., 2019*). Here, the raw data was searched against the mouse Swissprot database (release October 2017) using the paragon algorithm on Protein-Pilot (version 5.0.1, AB SCIEX). A total of 33,847 proteins were searched. As described (*Xu et al., 2019*), Bayesian protein-level differential quantification was performed by Andrew Dowsey (University of Bristol) using their own BayesProt (version 1.0.0), with default choice of priors and MCMC settings. Expression fold change relative to the control groups was determined and proteins with a global false discovery rate of <0.05 were deemed significant.

## Adipose tissue fractionation

Following a method adapted from *Collins et al., 2010*, gWAT was collected from adult male mice and washed in Hanks' Balanced Salt Solution (Sigma). Next, tissue was minced and digested in 1 mg/ml collagenase (Collagenase H, Sigma) for 30 min in a shaking incubator at 170 rpm, 37℃. The sample was then centrifuged at 1000 rpm for 5 min at 4℃. MA (floating layer) and stromal vascular fraction (SVF) (cell pellet) were collected separately, lysed in TRIzol Reagent (Invitrogen), and stored at −80℃ before proceeding to RNA extraction.

## 3T3-L1 cells

The 3T3-L1 cell line was purchased from ATCC (authentication and mycoplasma testing status as per ATCC documentation). Cells were maintained in Dulbecco's modified Eagle's medium (DMEM) – high glucose (D6429, Sigma-Aldrich) supplemented with 10% foetal bovine serum (FBS) and 1% penicillin/streptomycin (P/S) at 37℃/5% $CO_2$. Cells were grown until confluent, passaged and plated into 12-well tissue culture plates for differentiation. The differentiation protocol was initiated 5 days later. Cells were treated with 10 μg/ml insulin (Sigma-Aldrich), 1 μM dexamethasone (Sigma-Aldrich), 1 μM rosiglitazone (AdooQ Bioscience), and 0.5 mM IBMX (Sigma-Aldrich) prepared in DMEM + 10% FBS + 1% P/S for 3 days. On day 3 and day 5, the cell culture media was changed to 10 μg/ml insulin and 1 μM rosiglitazone in DMEM + 10% FBS + 1% P/S. On day 7, the cell culture media was changed to 10 μg/ml insulin in DMEM + 10% FBS + 1% P/S. Finally on day 10, the cell culture media was changed to DMEM + 10% FBS + 1% P/S without any additional differentiation mediators. Cells were used from day 11 onwards. Lipid droplets were visible by day 5.

For knockdown studies, mature 3T3-L1 adipocytes were transfected with SiControl (Control ON-TARGETplus siRNA, Dharmacon), SiNr1d1 (Mouse NR1D1 ON-TARGETplus siRNA, Dharmacon), or SiNr1d2 (Mouse NR1D2 ON-TARGETplus siRNA, Dharmacon) at 50 nM concentration using Lipofectamine RNAiMAX (Invitrogen) as a transfection reagent. Briefly, 12-well plates were coated with poly-L-lysine hydrobromide (Sigma) and incubated for 20–30 min prior to excess poly-L-lysine being removed and the plates allowed to dry. SiRNAs and RNAiMAX transfection reagent were separately mixed with reduced serum media (Opti-MEM, Gibco). The control or Nr1d1/β siRNA was then added to each well and mixed with an equal quantity of RNAiMAX and then incubated for 5 min at room temperature. Mature 3T3-L1 adipocytes were trypsinised (trypsin-EDTA solution, Sigma) and resuspended in FBS without P/S prior to being re-plated into the wells containing the SiRNA. After 24 hr the transfection mix was removed and replaced with DMEM without FBS or P/S. The cells were then collected 48 hr after transfection.

### RNA extraction (cells)

RNA was extracted from cells using the ReliaPrep RNA Cell Miniprep system (Promega UK), following manufacturer's instructions. RNA concentration and quality was determined with the use of a NanoDrop spectrophotometer and then stored at −80°C.

### RNA extraction (tissue)

Frozen adipose tissue was homogenised in TRIzol Reagent (Invitrogen) using Lysing Matrix D tubes (MP Biomedicals) and total RNA extracted according to the manufacturer's TRIzol protocol. To remove excess lipid, samples underwent an additional centrifugation (full speed, 5 min, room temperature) prior to chloroform addition. For the RNA sequencing samples, the isopropanol phase of TRIzol extraction was transferred to Reliaprep tissue Miniprep kit (Promega, Madison, WI) columns to ensure high-quality RNA samples were used. The column was then washed, DNAse treated, and RNA eluted as per protocol. RNA concentration and quality was determined with the use of a Nano-Drop spectrophotometer and then stored at −80°C. For RNA-seq, RNA was diluted to 1000 ng in nuclease-free water to a final volume of 20 μl.

### RNA extraction (adipose fractions)

MA and SVF were suspended in TRIzol Reagent (Invitrogen), and pipetted up and down to ensure cells were fully lysed. To remove excess lipid from MA fractions, samples were centrifuged (full speed, 5 min, room temperature) prior to chloroform addition. RNA extraction was then carried out as per the manufacturer's TRIzol protocol, up to the stage of removing the isopropranol phase, which was transferred to Reliaprep columns (Promega) for on-column DNase treatment, clean-up, and elution as per manufacturer's protocol.

### RT-qPCR

For RT-qPCR, samples were DNase-treated (RQ1 RNase-Free DNase, Promega, Madison, WI) prior to cDNA conversion High Capacity RNA-to-cDNA kit (Applied Biosystems). qPCR was performed using a GoTaq qPCR Master Mix (Promega, Madison, WI) and primers listed in Appendix Adipocyte NR1D1 dictates adipose tissue expansion during obesity using a Step One Plus (Applied Biosystems) qPCR machine. Relative quantities of gene expression were determined using the [delta][delta] Ct method and normalised with the use of a geometric mean of the housekeeping genes *Hprt*, *Ppib*, and *Actb* (housekeeping gene *Ppia* used for *Figure 4—figure supplement 1B,D*, *Figure 6—figure supplement 1D*). The fold difference of expression was calculated relative to the values of control groups.

### RNA-seq

Adipose tissue was collected from adult male mice (n=6–8/group) at ZT8 and flash-frozen. Total RNA was extracted and DNase-treated as described above. Biological replicates were taken forward individually to library preparation and sequencing. For library preparation, total RNA was submitted to the Genomic Technologies Core Facility (GTCF). Quality and integrity of the RNA samples were assessed using a 2200 TapeStation (Agilent Technologies) and then libraries generated using the TruSeq Stranded mRNA assay (Illumina, Inc) according to the manufacturer's protocol. Briefly, total RNA (0.1–4 μg) was used as input material from which polyadenylated mRNA was purified using poly-T, oligo-attached, magnetic beads. The mRNA was then fragmented using divalent cations under elevated temperature and then reverse-transcribed into first strand cDNA using random primers. Second strand cDNA was then synthesised using DNA Polymerase I and RNase H. Following a single 'A' base addition, adapters were ligated to the cDNA fragments, and the products then purified and enriched by PCR to create the final cDNA library. Adapter indices were used to multiplex libraries, which were pooled prior to cluster generation using a cBot instrument. The loaded flow cell was then paired-end sequenced (76 + 76 cycles, plus indices) on an Illumina HiSeq4000 instrument. Finally, the output data was demultiplexed (allowing one mismatch) and BCL-to-Fastq conversion performed using Illumina's bcl2fastq software, version 2.17.1.14.

## RNA-seq data processing and differential gene expression analysis

Paired-end RNA-seq reads were quality-assessed using FastQC (version 0.11.3), FastQ Screen (version 0.9.2) (*Wingett and Andrews, 2018*). Reads were processed with Trimmomatic (version 0.36) (*Bolger et al., 2014*) to remove any remaining sequencing adapters and poor quality bases. RNA-seq reads were then mapped against the reference genome (mm10) using STAR (version 2.5.3a) (*Dobin et al., 2013*). Counts per gene (exons) were calculated by STAR using the genome annotation from GENCODEM16. Differential expression analysis was then performed with edgeR (*Robinson et al., 2010*) using QLF tests based on published code (*Chen et al., 2016*). Changes were considered significant if they reached an FDR cut-off of <0.05. Transcripts with 'NA' gene ID are not included in gene numbers described. Interaction analysis was performed with stageR (*Van den Berge et al., 2017*) in conjunction with Limma voom (*Law et al., 2014*), setting alpha at 0.05.

## HaloChIP-seq

NR1D1 HaloChIP-seq was performed using a protocol based on *Hunter et al., 2020*, with modifications informed by *Castellano-Castillo et al., 2018*. Reagents were from the ChIP-IT High Sensitivity Chromatin Preparation kit (Active Motif 53046) and the HaloCHIP System (Promega G9410), unless otherwise specified. Freshly harvested gWAT was finely minced and dual-crosslinked in 10 ml 0.5 M disuccinimidyl glutarate (Thermo Fisher Scientific 20593) followed by 10 ml 1% formaldehyde-PBS. After quenching with glycine, tissue pieces were washed 2× in ice-cold ChIP wash buffer (Active Motif), then resuspended in 5 ml freshly made adipose lysis buffer (10 mM Tris HCl, 140 mM NaCl, 5 mM EDTA, 1% NP-40) supplemented with a HaloChIP-compatible protease inhibitor cocktail (Promega G6521). Fixation and wash solutions were changed by centrifuging tissue suspensions at 1250 *g* for 3 min (4°C), and removing the infranatant from below the floating adipose tissue pieces with glass Pasteur pipettes. Tissue was disrupted mechanically with the Qiagen TissueRuptor, then incubated on ice for 30 min to lyse cells. Following centrifugation (2780 *g* for 3 min, 4°C), the lipid layer and supernatant were carefully removed, and the nuclei pellet resuspended in 650 µl Mammalian Lysis Buffer (Promega), supplemented with protease inhibitor cocktail. Suspensions were once again incubated on ice (15 min) prior to low-amplitude probe sonication (Active Motif) (8× 2 min cycles of 30 s on/30 s off, 20% amplitude). Cellular debris was pelleted by spinning at 21,000 *g* for 3 min (4°C), and 600 µl of the chromatin suspension added to HaloLink Resin, prepared as per manufacturer's instructions. Pull-down reactions were rotated for 3 hr at room temperature, subsequent to wash and elution steps as per manufacturer's HaloCHIP System instructions. Eluted ChIP DNA was purified with the ChIP DNA Clean and Concentrator kit (Zymo D5205). ChIP DNA was quantified with the Qubit dsDNA HS Assay Kit (Invitrogen Q32851). For experimental samples, tissue was collected from male homozygous *HaloNr1d1* mice at ZT8; for control samples, tissue was collected from male homozygous *HaloNr1d1* mice at ZT20, and from male WT mice (related colony) at ZT8. For each ChIP-seq library preparation, ChIP DNA from three to seven mice was pooled to total 5 ng ChIP DNA. ChIP DNA pools (2× ZT8 *HaloNr1d1* pools, 1× ZT20 *HaloNr1d1* pool, 1× ZT8 WT pool) were submitted to the University of Manchester GTCF for TruSeq ChIP library preparation (Illumina) as per manufacturer's instructions and paired-end sequencing (HiSeq 4000).

## HaloChIP-seq data processing

Adapter trimming was performed with Trimmomatic v0.38.1 (5) using Galaxy version 20.01, specifying the following parameters: ILLUMINACLIP:TruSeq3-PE.fa:2:30:10 LEADING:3 TRAILING:3 SLIDINGWINDOW:4:20 MINLEN:36. Paired reads were aligned to the mm10 genome with Bowtie 2 v2.3.4.3+galaxy0 (*Langmead and Salzberg, 2012*) using default settings. Duplicates were removed with Picard v2.18.2.1 (Broad Institute) with –REMOVE_DUPLICATES = true. Resulting mean library size was 89.5M reads.

## HaloChIP-seq data analysis

Peaks were called using MACS2 v2.2.7.1 (*Zhang et al., 2008*), using: macs2 callpeak –t BAM_FILES –c BAM_FILES –g mm –f BAMPE –q 0.01. Peaks were called from the ZT8 *HaloNr1d1* libraries using either ZT8 WT or ZT20 *HaloNr1d1* libraries as controls. Peaks commonly called by both strategies were identified using bedtools v2.19.1 (*Quinlan, 2014*) intersect tool. Motif analysis was performed

using HOMER v4.9.1 (*Heinz et al., 2010*), running: findMotifsGenome.pl BED_FILE mm10 –size 200 –mask ChIP data was visualised with Integrative Genomics Viewer v2.4.6 (*Thorvaldsdóttir et al., 2013*).

## Integrating RNA-seq and ChIP-seq

In order to calculate enrichment of RNA-seq-based gene clusters with respect to ChIP-seq peaks, we used our in-house custom tool (*Briggs et al., 2021b*; *Yang et al., 2019*) which calculates gene cluster enrichment within specified distances from the centre of peaks (see also *Code Availability* statement below). This Peak-set Enrichment of Gene-sets (PEGS) tool extends peaks in both directions for the given distances and extracts all genes whose TSSs overlap with the extended peaks. Given these genes, the inputted RNA-seq-based gene cluster, and the overlap of these two groups, it performs a hypergeometric test with the total number of genes in the mm10 genomes as background. Parameters were set as: pegs refGene mm10_intervals.bed BED_FILES/ GENE_LISTS/ 100 500 1000 5000 10000 50000 100000 500000 1000000 5000000 10000000.

## Pathway analysis

Pathway enrichment analysis of gene identifiers, either extracted from RNA-seq or proteomics data, was carried out using IPA (Qiagen) as described in *Krämer et al., 2014*, or using the R Bioconductor package ReactomePA (*Yu and He, 2016*). For ReactomePA, the *enrichPathway* tool was used with the following parameters: organism = 'mouse', pAdjustMethod = 'BH', maxGSSize = 2000, readable = FALSE. We considered pathways with a padj<0.01 to be significantly enriched.

## Protein extraction and Western blotting

Small pieces (<100 mg) of tissue were homogenised with the FastPrep Lysing Matrix D system (MP Biomedicals) in T-PER (Thermo Fisher Scientific), supplemented with protease inhibitor cocktail (Promega) at 1:50 dilution. Benzonase nuclease (EMD Millipore) was added (2 µl), the homogenate briefly vortexed, then incubated on ice for 10 min. Homogenates were then centrifuged for 8 min at 10,000 *g*, at 4°C, and the supernatant removed (avoiding any lipid layer). Protein concentration was quantified using the Bio-Rad Protein Assay (Bio-Rad). For Western blotting, equal quantities (75 µg for detection of NR1D1) of protein were added to 4× NuPAGE LDS sample buffer (Invitrogen), NuPAGE sample reducing agent (dithiothreitol) (Invitrogen) and water, and denatured at 70°C for 10 min. Samples were run on 4–20% Mini-PROTEAN TGX Precast Protein Gels (Bio-Rad) before wet transfer to nitrocellulose membranes. Membranes were blocked with Protein-Free Blot Blocking Buffer (Azure Biosystems), and subsequent incubation and wash steps carried out following manufacturer's instructions. Primary and secondary antibodies used were as listed in the Key resources table, with primary antibodies being used at a 1:1000 dilution and secondary antibodies at 1:10,000. Membranes were imaged using chemiluminescence or the LI-COR Odyssey system. Uncropped blot images are provided in the Source Data Files.

## Statistics

To compare two or more groups, t-tests or ANOVAs were carried out using GraphPad Prism (v.8.4.0). For all of these, the exact statistical test used and n numbers are indicated in the figure legends. All n numbers refer to individual biological replicates (i.e. individual animals). Cell line experiments were replicated three times; animal experiments were typically performed only once. Notable exceptions included two independent assessments of HFD feeding in the *Nr1d1^{Flox2-6}:Adipoq^{Cre}* mice, and the two independent RNA-seq experiments comparing *Nr1d1^{Flox2-6}:Adipoq^{Cre}* Cre^{-ve} and Cre^{+ve} mice on NC (*Figure 3C*, *Figure 5B*). Unless otherwise specified, bar height is at mean, with error bars indicating ± SEM. In these tests, significance is defined as *p<0.05 or **p<0.01 (p-values below 0.01 were not categorised separately, i.e. no more than two stars were used, as we deemed this to be a meaningful significance cut-off). Statistical analyses of proteomics, RNA-seq, and ChIP-seq data were carried out as described above in Materials and methods, using the significance cut-offs mentioned. Plots were produced using GraphPad Prism or R package ggplot2.

## Code availability

The custom Python code (*Briggs et al., 2021b*; *Yang et al., 2019*) used to carry out the peaks-genes enrichment analysis in this study is available as the PEGS package at https://github.com/fls-bioinformatics-core/pegs (*Briggs, 2021a*) and through the Python Package Index (PyPI).

## Acknowledgements

We thank Rachel Scholey, I-Hsuan Lin, Ping Wang, and Peter Briggs (Bioinformatics Core Facility, UoM), Emma Smith and Thea Danby (Faculty of Biology, Medicine and Health, UoM) for statistical and technical assistance, and acknowledge support of core facilities at the University of Manchester: Genomic Technologies Core Facility, Biological Services Unit, and Histological Services Unit. We thank Argentina-Gabriela Baluta (Faculty of Biology, Medicine and Health, UoM) for useful insights into the RNA-seq data. We also acknowledge and thank the support of our funders: the BBSRC (BB/I018654/1 to DAB), the MRC (Clinical Research Training Fellowship MR/N021479/1 to ALH; MR/P00279X/1 to DAB; MR/P011853/1 and MR/P023576/1 to DWR), and the Wellcome Trust (107849/Z/15/Z, 107851/Z/15/Z).

## Additional information

### Competing interests

Jean-Noel Billaud: J-N.B. is an employee of Qiagen. The other authors declare that no competing interests exist.

### Funding

| Funder | Grant reference number | Author |
| --- | --- | --- |
| Biotechnology and Biological Sciences Research Council | BB/I018654/1 | David A Bechtold |
| Medical Research Council | MR/N021479/1 | Ann Louise Hunter |
| Medical Research Council | MR/P00279X/1 | David A Bechtold |
| Medical Research Council | MR/P011853/1 | David W Ray |
| Medical Research Council | MR/P023576/1 | David W Ray |
| Wellcome Trust | 107849/Z/15/Z | David W Ray |
| Wellcome Trust | 107851/Z/15/Z | David W Ray |

The funders had no role in study design, data collection and interpretation, or the decision to submit the work for publication.

### Author contributions

Ann Louise Hunter, Conceptualization, Software, Formal analysis, Funding acquisition, Investigation, Writing - original draft; Charlotte E Pelekanou, Conceptualization, Formal analysis, Investigation, Writing - original draft; Nichola J Barron, Rebecca C Northeast, Magdalena Grudzien, Polly Downton, Thomas Cornfield, Peter S Cunningham, Investigation; Antony D Adamson, Methodology; Jean-Noel Billaud, Leanne Hodson, Formal analysis; Andrew SI Loudon, Writing - review and editing; Richard D Unwin, Formal analysis, Methodology; Mudassar Iqbal, Software, Formal analysis; David W Ray, Conceptualization, Funding acquisition, Methodology, Writing - original draft, Writing - review and editing; David A Bechtold, Conceptualization, Formal analysis, Supervision, Funding acquisition, Methodology, Writing - original draft, Project administration, Writing - review and editing

### Author ORCIDs

Ann Louise Hunter (iD) https://orcid.org/0000-0002-3874-4852
Rebecca C Northeast (iD) http://orcid.org/0000-0002-3121-2802

Polly Downton 🄳 http://orcid.org/0000-0002-1617-6153
David A Bechtold 🄳 https://orcid.org/0000-0001-8676-8704

### Ethics

Animal experimentation: All experiments described here were conducted in accordance with local requirements and licenced under the UK Animals (Scientific Procedures) Act 1986, project licence number 70/8558 (licence holder Dr. David A Bechtold). Procedures were approved by the University of Manchester Animal Welfare and Ethical Review Body (AWERB).

### Decision letter and Author response

Decision letter https://doi.org/10.7554/eLife.63324.sa1
Author response https://doi.org/10.7554/eLife.63324.sa2

# Additional files

## Supplementary files

• Transparent reporting form

## Data availability

RNA-seq data generated in the course of this study has been uploaded to ArrayExpress and is available at http://www.ebi.ac.uk/arrayexpress/experiments/E-MTAB-8840. ChIP-seq data generated in the course of this study has been uploaded to ArrayExpress and is available at http://www.ebi.ac.uk/arrayexpress/experiments/E-MTAB-10573. Raw proteomics data has been uploaded to Mendeley Data at https://data.mendeley.com/datasets/wskyz3rhsg/draft?a=ef40a1ec-36a4-4509-979d-32d494b96585. Output of 'omics analyses (proteomics, edgeR, stageR, ReactomePA outputs, peak calling) are provided in the Source Data Files.

The following datasets were generated:

| Author(s) | Year | Dataset title | Dataset URL | Database and Identifier |
|---|---|---|---|---|
| Hunter AL, Pelekanou CE, Barron NJ, Northeast RC, Grudzien M, Adamson AD, Downton P, Cornfield T, Cunningham PS, Billaud JN, Hodson L, Loudon A, Unwin RD, Iqbal M, Ray D, Bechtold DA | 2021 | Adipocyte NR1D1 dictates adipose tissue expansion during obesity - RNA-seq | http://www.ebi.ac.uk/arrayexpress/experiments/E-MTAB-8840 | ArrayExpress, E-MTAB-8840 |
| Hunter AL, Pelekanou CE, Barron NJ, Northeast RC, Grudzien M, Adamson AD, Downton P, Cornfield T, Cunningham PS, Billaud JN, Hodson L, Loudon A, Unwin RD, Iqbal M, Ray D, Bechtold DA | 2021 | Adipocyte NR1D1 dictates adipose tissue expansion during obesity | http://dx.doi.org/10.17632/wskyz3rhsg.4 | Mendeley Data, 10.17632/wskyz3rhsg.4 |
| Hunter AL, Pelekanou CE, Barron NJ, Northeast RC, Grudzien M, | 2021 | Adipocyte NR1D1 dictates adipose tissue expansion during obesity - ChIP-seq | http://www.ebi.ac.uk/arrayexpress/experiments/E-MTAB-10573 | ArrayExpress, E-MTAB-10573 |

Adamson AD,
Downton P,
Cornfield T,
Cunningham PS,
Billaud JN, Hodson
L, Loudon A, Unwin
RD, Iqbal M, Ray D,
Bechtold DA

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

# Appendix 1

## qPCR primer sequences

| Gene | Forward primer (5′–3′) | Reverse primer (5′–3′) |
| --- | --- | --- |
| Acaca | TAATGGGCTGCTTCTGTGACTC | TCAATATCGCCATCACTCTTG |
| Acsl1 | TGGGGTGGAAATCATCAGCC | CACAGCATTACACACTGTACAACGG |
| Acss3 | AATGTCGCAAAGTAACAGGCG | GTGGGTCTTGTACTCACCACC |
| Actb | GGCTGTATTCCCCTCCATCG | CCAGTTGGTAACAATGCCATGT |
| Aldoa | CGTGTGAATCCCTGCATTGG | CAGCCCCTGGGTAGTTGTC |
| Arntl | GTCGAATGATTGCCGAGGAA | GGGAGGCGTACTTGTGATGTTC |
| Cd36 | CCACAGTTGGTGTGTTTTATCC | TCAATTATGGCAACTTTGCTT |
| Col1a1 | TCCCAGAACATCACCTATCAC | CTGTTGCCTTCGCCTCTGAG |
| Col5a3 | TACCTCTGGTAACCGGGGTCTC | CCTTTTGGTCCCTCATCACCC |
| Col6a1 | TGCCCTGTGGATCTATTCTTCG | CTGTCTCTCAGGTTGTCAATG |
| Col6a2 | TGGTCAACAGGCTAGGTGCCAT | TAGACAGGGAGTTGACTCGCTC |
| Col6a3 | CTGTCGCCTGCATTCATC | ACAACCCTCTGCACAAAGTC |
| Cs | TGACTGGCACCCAACATTTGA | CAGCTTGAGGCACAGCAGGTATAG |
| Dio2 | CCAGACAACTAGCATGGCGT | GAAAATTGGCTGCCCCACAC |
| Elovl6 | GAGCAGAGGCGCAGAGAAC | ATGCCGACCACCAAAGATAA |
| Fasn | CCCAGAGGCTTGTGCTGACT | CGAATGTGCTTGGCTTGGT |
| G6pdx | AGACCTGCATGAGTCAGACG | TGGTTCGACAGTTGATTGGA |
| Hprt | GTTGGATACAGGCCAGACTTTGTTG | GATTCAACTTGCGCTCATCTTAGGC |
| Loxl4 | TTGCTCTCAAGGACACCTGGTA | GCAGCGAACTCCACTCATCA |
| Lpl | AGGGCTCTGCCTGAGTTGTA | CCATCCTCAGTCCCAGAAAA |
| Me1 | GGGATTGCTCACTTGGTTGT | GTTCATGGGCAAACACCTCT |
| Pfk1 | TGCAGCCTACAATCTGCTCC | GTCAAGTGTGCGTAGTTCTGA |
| Plin2 | AAGAGGCCAAACAAAAGAGCCAGGAGACCA | ACCCTGAATTTTCTGGTTGGCACTGTGCAT |
| Pnpla2 | TGTGGCCTCATTCCTCCTAC | TCGTGGATGTTGGTGGAGCT |
| Ppia | TATCTGCACTGCCAAGACTGAGTG | CTTCTTGCTGGTCTTGCCATTCC |
| Ppib | GGAGATGGCACAGGAGGAAA | CCGTAGTGCTTCAGTTTGAAGTTCT |
| Nr1d1 | GTCTCTCCGTTGGCATGTCT | CCAAGTTCATGGCGCTCT |
| Nr1d2 | CAGGAGGTGTGATTGCCTACA | GGACGAGGACTGGAAGCTAAT |
| Scd1 | CGCTGGTGCCCTGGTACTGC | CAGCCAGGTGGCGTTGAGCA |
| Ucp1 | ACTGCCACACCTCCAGTCATT | CTTTGCCTCACTCAGGATTGG |

