## [Decision Letter]

**Acceptance summary:**

This manuscript clarifies the role of NR1D1/REVERBa in transcriptional regulation in adipose tissue. The authors show that the phenotype resulting from adipose NRD1 knockout is distinct to what was previously reported in a global knockout model. The findings reveal that loss of adipose REVERBa leads to healthy adipose expandability with systemic metabolic benefits.

**Decision letter after peer review:**

Thank you for submitting your article "Adipocyte REVERBα dictates adipose tissue expansion during obesity" for consideration by *eLife*. Your article has been reviewed by 3 peer reviewers, one of whom is a member of our Board of Reviewing Editors, and the evaluation has been overseen by David James as the Senior Editor. The reviewers have opted to remain anonymous.

The reviewers have discussed the reviews with one another and the Reviewing Editor has drafted this decision to help you prepare a revised submission.

Summary:

The manuscript by Hunter et al., identifies a dominant metabolic role for REVERBa in adipose tissue. By using an adipose-specific REVERBa knockout mouse model, the authors describe a metabolic phenotype distinct from that previously reported in a global model. Convincing data is presented supporting a role for REVERBa in regulating the circadian clock and collagen dynamics, the latter of which was not previously recognized. The authors further show that loss of adipose REVERBa leads to healthy adipose expandability with added systemic metabolic benefit. Overall, the results supporting a broader role for REVERBa to include lipid and mitochondrial metabolic pathways in the context of obesity are of interest. However, the reviewers also identified a number of areas where the current data do not fully support the conclusions drawn.

Essential revisions:

1) The authors are encouraged to provide mechanistic insights detailing altered REVERBa repressive activity upon high fat-feeding. Is activity of REVERBa physiologically regulated in response to HFD, e.g., by altered post-translational modifications and/or by altered co-repressor complex interactions? Also, is adipose REVERBa expression by itself affected during obesity?

2) Prior publications have shown that the metabolic gene regulation mediated by REVERBa tends to also involve HDAC3. They should consider further mechanistic studies of REVERBa and HDAC3 in this mouse line.

3) One major issue with this study is the lack of cellular specificity. The authors generate mice with an adipocyte-specific knockout of REVERBa, but then do RNA-seq of the entire gonadal fat pad. Since only about 50% of the cells in the tissue are adipocytes, there is thus no way to know whether the gene expression changes seen are due to direct loss of REVERBa in adipocytes or secondary effects on other cell types within adipose tissue. The authors could partly address this concern by either (a) fractionating adipose tissue into mature adipocytes and stromal vascular fraction to verify which cell types are demonstrating the expression changes or (b) isolating SVF from control and adipocyte-specific knockout animals and differentiating them in vitro to determine cell intrinsic effects of REVERBa deletion.

4) The lack of specificity also becomes an issue with the analyses done in Figure 6 in which they have overlaid their RNA-Seq data with previously published ChIP-seq data. They then use bioinformatics to predict which genes are direct targets of REVERBa. This analysis would be strengthened if they first knew which of their transcriptional changes represented expression differences in adipocytes and if they then either did their own ChIP-seq study or at least validated some candidate genes by ChIP-PCR to actually confirm REVERBa binding.

5) In Figure 3, the authors highlight that adipocyte-specific knockout of REVERBa results in increased collagen gene expression (Figure 3). However, on a high fat diet, mutant animals actually have decreased fibrosis. How do the authors explain this seeming contradiction? At a minimum, they should measure the same panel of collagen genes they propose to be regulated by REVERBa to see if they are altered in high fat fed gWAT.

6) Some of the claims made on the metabolic phenotyping are overstatements:

a) The data presented on acute cold exposure and REVERBb compensation in 3T3-L1 adipocytes are not at all sufficient to claim that the phenotypes seen here are not due to effects in brown fat or due to compensatory action by REVERBb.

b) Isolated sections with F4/80 alone are not sufficient to make a broad claim about inflammation. This would be more convincing if supported by flow cytometry data and inflammatory gene expression.

c) The legend for Figure 4 says that mutant mice have "greater insulin sensitivity". However, the curves in Figure 4F look very similar. Moreover, they calculate the area within the curve. I am not sure if they mean the area under the curve, which is more typically done, but this is confusing and needs to be clarified.

---

## [Author Response]

Summary:The manuscript by Hunter et al., identifies a dominant metabolic role for REVERBa in adipose tissue. By using an adipose-specific REVERBa knockout mouse model, the authors describe a metabolic phenotype distinct from that previously reported in a global model. Convincing data is presented supporting a role for REVERBa in regulating the circadian clock and collagen dynamics, the latter of which was not previously recognized. The authors further show that loss of adipose REVERBa leads to healthy adipose expandability with added systemic metabolic benefit. Overall, the results supporting a broader role for REVERBa to include lipid and mitochondrial metabolic pathways in the context of obesity are of interest. However, the reviewers also identified a number of areas where the current data do not fully support the conclusions drawn.

We thank the reviewers and editors for their positive feedback on our work. As outlined below, we have added new data, analyses, and discussion to further support the main conclusions of the paper. Specifically, we have now undertaken and present: (i) HaloNR1D1 ChIP-seq studies; (ii) assessment of gene expression in mature adipocytes isolated from gWAT of NC- and HFD-fed mice; (iii) additional characterisation of energy expenditure in *Nr1d1^Flox2-6^:Adipoq^Cre^* and *Nr1d1^Flox2-6^
*mice; (iv) additional characterisation of immune/inflammation-related processes in the HFD-fed mice, and (v) extended analyses of relevant published datasets.

Overall, our studies demonstrate that NR1D1 (REVERBα) is highly influential over the adipose tissue response to high-fat diet (HFD) feeding, and the development of obesity-related pathology. Underlying this is a repression of metabolic function in response to chronic over-nutrition, which is mitigated upon genetic targeting of *Nr1d1*. Integration of RNA-seq and ChIP-seq studies indicates that these effects are due in large part to direct repressive action of NR1D1. Metabolic genes which are shut down in obese adipose show significant enrichment at sites of NR1D1 chromatin binding, and are significantly up-regulated in *Nr1d1^Flox2-6^:Adipoq^Cre^* adipose. A similar proximity relationship was not observed for genes showing decreased expression in the *Nr1d1* targeted mice, in line with the constitutive repressive function of NR1D1 (and supporting the specificity of our analyses).

In contrast, the impact of adipose *Nr1d1* deletion on gene expression and phenotype was minimal under normal chow feeding conditions, despite a wide-reaching cistrome (as demonstrated by our adipose HaloNR1D1 ChIP-seq). Thus, NR1D1 has the potential to regulate a broad programme of gene expression, linked to widespread chromatin binding, but this does not occur in the basal state. It is only revealed under a state of obesity.

Essential revisions:1) The authors are encouraged to provide mechanistic insights detailing altered REVERBa repressive activity upon high fat-feeding. Is activity of REVERBa physiologically regulated in response to HFD, e.g., by altered post-translational modifications and/or by altered co-repressor complex interactions? Also, is adipose REVERBa expression by itself affected during obesity?

We understand the reviewers/editors’ request for additional consideration of how NR1D1 (REVERBα) delivers obesity-dependent transcriptional control over metabolic processes. To this end, we have undertaken additional studies, including the generation of a novel, antibody-independent NR1D1 cistrome in adipose tissue, transcriptional analyses in mature adipocytes under normal and obese conditions, assessment of NR1D1 expression under chow and HFD-fed conditions, and additional integration with published literature (these additions are discussed in detail below). Overall, these data suggest that altered NR1D1 expression and/or interaction with co-repressors *per se* is unlikely to explain the emergence of increased repressive influence by NR1D1 under conditions of obesity.

NR1D1 expression. In our studies, we find a large effect of NR1D1 loss in obese adipose, which we do not see in the basal state. This is important, as it suggests that NR1D1 repressive activity increases in obesity. However, as we and others have shown, *Nr1d1* gene expression is consistently decreased in white adipose tissue following long-term HFD feeding in mice (e.g. Cunningham et al., 2016). This was similarly observed in our RNA-seq studies in the current work (*Nr1d1* expression showing log fold-change of -1.56 between HFD- and NC-fed *Nr1d1^Flox2-6^* adipose, FDR 1.19E-06). Therefore, increased *Nr1d1* gene expression cannot explain the increased influence of the nuclear receptor in the obese state. To examine whether there is a similar impact of chronic HFD-feeding on NR1D1 protein expression, we undertook Western blot analyses following 16 weeks of NC or HFD-feeding (see Author response image 1). Here, we similarly find no evidence of increased protein expression in obese gWAT. We therefore have no evidence that NR1D1 is expressed more highly in WAT of obese animal.

**Author response image 1. respfig1:** Western blot (top panel) showing NR1D1 protein expression in NC and HFD gWAT. Corresponding Ponceau stain (bottom panel).

Altered mode of action. DNA binding and subsequent repressive activity of NR1D1 involves two key mechanisms: firstly by competing with the ROR transcriptional activators at ROR-elements (ROREs; Forman et al., 1994), and secondly by recruiting co-repressors NCOR and HDAC3 to sites of NR1D1 binding at RevDR2 elements or co-located RORE motifs (Phelan et al., 2010; Zamir et al., 1996, 1997). Our new HaloNR1D1 cistrome reveals widespread chromatin binding in gWAT under basal state conditions (i.e. NC-fed mice), and significant enrichment of both RORE and RevDR2 motifs at NR1D1 binding sites. Classic examples of these two repressive mechanisms include clock genes *Arntl (Bmal1)* and *Nfil3*. NR1D1 repression of the core clock gene *Arntl* is mediated through the binding of NR1D1 to two closely situated RORE motifs in the *Arntl* promoter (Preitner et al., 2002; Yin and Lazar, 2005). By contrast, NR1D1 repression of clock gene *Nfil3* (*E4bp4*) likely occurs by antagonization of ROR transactivation, with NR1D1 binding a promoter-located RORE motif as a monomer (Duez et al., 2008). In our studies, we observe significant de-repression of both *Arntl* and *Nfil3* under both NC and obese conditions (Author response image 2). This suggests that both modes of NR1D1 repressive activity are evident in the basal (NC) state, and does not support an alteration in the functioning of NR1D1 in the obese state. Whilst it would be of interest to investigate whether there are more subtle differences in the NR1D1 protein under normal and obese conditions, studying the native NR1D1 protein is very challenging due to poor sensitivity of NR1D1 antibodies. We believe that such studies are beyond the scope of this current work.

**Author response image 2. respfig2:** Gene expression (mean +/- SEM of raw counts) of core clock genes Arntl and Nfil3 in bulk gWAT RNA-seq from NC- or HFD-fed Nr1d1^Flox2-6^:Adipoq^Cre^ mice and Nr1d1^Flox2-6^controls. A significant difference (FDR<0.05) between genotypes is detected for each comparison. N=4-6/group.

We can, however, gain mechanistic insight by examining the genome-wide DNA binding profile of NR1D1. The initial version of our manuscript made use of an existing NR1D1 cistrome (taken from Zhang et al., 2015), with stringent peak-calling parameters applied. Stringent criteria were applied because, as we have shown ourselves, overlap between antibody-dependent NR1D1 cistromes is limited, likely reflecting poor antibody sensitivity and specificity (Hunter et al., 2020). However, we have now been able to profile the gWAT NR1D1 cistrome in an antibody-independent manner, and thus substantially add to this study. We have used *HaloNr1d1* mice which express (at the endogenous *Nr1d1* locus) NR1D1 protein with a HaloTag fused to its N-terminus. HaloNR1D1 demonstrates normal DNA-binding domain function (no de-repression of NR1D1 target genes is seen in *HaloNr1d1* tissue), and *HaloNr1d1* mice show normal fertility, behaviour, and body composition (Hunter et al., 2020). By performing HaloNR1D1 ChIP-seq in gWAT collected from *HaloNr1d1* mice at ZT8 (peak NR1D1 genomic recruitment), and calling peaks against two control libraries (WT ZT8, *HaloNr1d1* ZT20), we mapped a NR1D1 cistrome of over 4,000 binding sites (Figure 6A).

Therefore, under basal conditions, the NR1D1 cistrome is far broader than the limited effects of *Nr1d1* deletion would suggest. To support a functional role for this broad cistrome, we looked for evidence that the cistrome correlates with NR1D1 regulation of gene expression (Figure 6B,C). We examined the relationship between NR1D1 binding sites and the genes affected by *Nr1d1* deletion, under both NC and HFD conditions, and looked for associations that are more significant than would occur by chance (i.e. that might occur for any gene in the genome). We did not see any association between NR1D1 binding sites and genes down-regulated with NR1D1 loss, in keeping with NR1D1’s constitutive repressor function, and supporting the specificity of our analysis. Importantly, we did see strong associations between NR1D1 binding sites and genes up-regulated with NR1D1 loss, not only in the basal (NC-fed) state, but under HFD conditions also. This suggests that NR1D1 has the potential to regulate a broad programme of gene expression, but this does not occur in the basal state. We propose that, in the basal state, chromatin organisation and the action of other transcriptional regulators combine to make NR1D1 repressive activity redundant. In the perturbed, obese state, rewiring of the regulatory landscape serves to make NR1D1 repression important. Exploration of this hypothesis is intended for ongoing and future studies. There is supportive data, however, from studies which demonstrate chromatin alterations in obese adipose (Roh et al., 2020; Zhang et al., 2018), and studies in mouse liver which have shown remodelling of the gene regulatory environment in response to high-fat diet (Dyar et al., 2018; Quagliarini et al., 2019).

2) Prior publications have shown that the metabolic gene regulation mediated by REVERBa tends to also involve HDAC3. They should consider further mechanistic studies of REVERBa and HDAC3 in this mouse line.

One of the mechanisms of NR1D1-mediated repression is indeed recruitment of the NCOR/HDAC3 co-repressor complex. An earlier report suggested that NR1D1 represses distinct programmes of genes (most notably core clock-related versus metabolic genes) through different mechanisms (Zhang et al., 2015). Specifically, this report suggested that NR1D1 repression of clock genes involved direct binding to DNA, whereas repression of metabolic genes was accomplished by tethering to lineage-determining factors (e.g. HNF6 in liver). This theory stemmed from experiments performed in the *Nr1d1^DBDm^* mouse, a conditional transgenic model expressing a DNA-binding domain mutant form of NR1D1. We have shown that the *Nr1d1^DBDm^* mouse is a phenocopy of the *Nr1d1^-/-^* mouse (Hunter et al., 2020); ie. it is not spared the metabolic abnormalities observed in the global knockout model. Furthermore, in HaloNR1D1 ChIP-seq in liver, there is no enrichment of lineage-determining factor motifs above and beyond what would be expected at any sites of open chromatin (Hunter et al., 2020). As we now also observe in our adipose HaloNR1D1 ChIP-seq (employing dual-crosslinking to maximise capture of any tethered interactions), the RORE and RevDR2 motifs are by far the dominant transcription factor binding motifs underlying NR1D1 DNA binding. Therefore, NR1D1 repression requires DNA-binding, and as such, may be mediated through competition with RORs, not only through NCOR/HDAC3 recruitment.

Existing evidence suggests that NCOR and HDAC3 activity extends far beyond NR1D1 action (Feng et al., 2011; Ferrari et al., 2017; Li et al., 2011), with these factors having wide-ranging repressive activity, including of critical metabolic regulators such as PPARγ. In further support is our report that, in liver, we find substantial overlap between NR1D1, HDAC3, and NCOR cistromes, but also large numbers of sites bound by only one of these factors (Hunter et al., 2020). We have been interested to examine HDAC3 activity in adipose, but HDAC3 does not directly bind the genome, thus HDAC3 ChIP has proven challenging (we attempted this in response to reviewer/editor feedback, but without success). We can use removal of the H3K27ac mark as a proxy measure of HDAC3 activity however (Nguyen et al., 2020). Here, we do see removal of the H3K27ac mark at NR1D1 binding sites (adipocyte H3K27ac ChIP-seq data taken from Roh et al., 2020) under both NC and HFD-feeding conditions (Author response image 3), at both established NR1D1 targets, and at NR1D1 targets unmasked by obesity. This suggests that altered interaction between NR1D1 and HDAC3/NCOR is not the driving factor in increased NR1D1 repression in obese mice, although we cannot exclude this possibility with current data. Given that NR1D1 does not mediate gene repression exclusively through HDAC3, and that HDAC3 does not interact exclusively with NR1D1, we believe that pursuing extensive, technically difficult studies of HDAC3-NR1D1 interaction state under different chronic feeding conditions is beyond the scope of the current manuscript.

**Author response image 3. respfig3:** Adipocyte H3K27ac ChIP-seq signal, collected in NC (top row) and HFD (middle row) conditions, aligned with adipose HaloNR1D1 ChIP-seq signal (bottom row).

3) One major issue with this study is the lack of cellular specificity. The authors generate mice with an adipocyte-specific knockout of REVERBa, but then do RNA-seq of the entire gonadal fat pad. Since only about 50% of the cells in the tissue are adipocytes, there is thus no way to know whether the gene expression changes seen are due to direct loss of REVERBa in adipocytes or secondary effects on other cell types within adipose tissue. The authors could partly address this concern by either (a) fractionating adipose tissue into mature adipocytes and stromal vascular fraction to verify which cell types are demonstrating the expression changes or (b) isolating SVF from control and adipocyte-specific knockout animals and differentiating them in vitro to determine cell intrinsic effects of REVERBa deletion.

We recognise that this is a critical question, and have undertaken additional experiments and analyses to address this. Undoubtedly, gene expression changes observed on bulk analyses reflect effects across all adipose cell types. Therefore, as suggested by the reviewers, we have now completed an adipose tissue fractionation study, in which *Nr1d1^Flox2-6^:Adipoq^Cre^* and *Nr1d1^Flox2-6^
*mice were maintained on HFD or NC for 16 weeks, with subsequent isolation of mature adipocytes from freshly-collected gWAT. This method (adapted from Collins et al., 2010) clearly separated a population of adiponectin (*Adipoq*)-expressing mature adipocytes from the remaining stromal vascular fraction. These studies (now included in Figure 6 Supplemental) provide strong support for the conclusions drawn from the whole tissue RNA-seq (regarding lack of differential gene expression under NC conditions, and unmasking of differential gene expression with chronic HFD feeding in the adipocyte knockout mice). Specifically, we assessed expression of 7 genes, all of which show up-regulation in *Nr1d1^Flox2-6^:Adipoq^Cre^*gWAT under HFD but not NC conditions in bulk RNA-seq, and which all have a TSS within 100kbp of a NR1D1 ChIP-seq peak (Figure 6SD). In line with our previous studies, we saw no between-genotype differences in any of these genes in mature adipocytes isolated from NC-fed mice. In contrast, mature adipocytes isolated from HFD-fed mice showed a pronounced up-regulation in *Nr1d1^Flox2-6^:Adipoq^Cre^* samples (compared to control HFD-fed mice), strongly suggesting that there is an effect specific to adipocyte gene expression with adipocyte-targeted *Nr1d1* deletion. These results are described on lines 308-311 of the main manuscript.

Recent data from the literature also reveal a major impact of chronic HFD-feeding on the adipocyte transcriptome. In their recent study, Evan Rosen’s group (Roh et al., 2020) used the NuTRAP reporter to profile adipocyte-specific gene expression and H3K27 acetylation in NC and HFD-fed mice. We have analysed the raw RNA-seq data from this paper and compared it with our bulk adipose RNA-seq data. In line with our original conclusions, genes commonly (adipocyte and whole tissue) up-regulated by HFD-feeding are strongly enriched for immune pathways, whilst down-regulated genes enrich for lipid and mitochondrial metabolism pathways (Author response table 1). Overall, these data reinforce our earlier conclusions that metabolic pathways are shut down in adipocytes in response to long-term HFD feeding, and that genetic targeting of *Nr1d1* expression in adipocytes mitigates this effect.

**Author response table 1. resptable1:** ReactomePA pathway analysis for genes commonly dysregulated by HFD (FDR<0.05, HFD vs NC) in our study, and an adipocyte-only RNA-seq study.

Top 5 ReactomePA pathways (by gene count) for 749 genes up-regulated by HFD in both our data (Cre-) and Roh et al. adipocyte-only RNA-seq:					
ID	Description	GeneRatio	BgRatio	p.adjust	Count
R-MMU-168256	Immune System	463/438	1691/8733	1.90E-16	163
R-MMU-168249	Innate Immune System	108/438	911/8733	5.32E-16	108
R-MMU-109582	Hemostasis	70/438	505/8733	4.92E-13	70
R-MMU-1280218	Adaptive Immune System	65/438	667/8733	5.89E-06	65
R-MMU-6798695	Neutrophil degranulation	62/438	514/8733	6.32E-09	62
Top 5 ReactomePA pathways (by gene count) for 826 genes down-regulated by HFD in both our data (Cre-) and Roh et al., adipocyte-only RNA-seq:					
ID	Description	GeneRatio	BgRatio	p.adjust	Count
R-MMU-1430728	Metabolism	177/421	1727/8733	4.85E-24	177
R-MMU-556833	Metabolism of lipids	52/421	594/8733	0.000992	52
R-MMU-1428517	The citric acid (TCA) cycle and respiratory electron transport	37/421	143/8733	3.61E-15	37
R-MMU-71291	Metabolism of amino acids and derivatives	33/421	247/8733	9.30E-06	33
R-MMU-8978868	Fatty acid metabolism	25/421	172/8733	5.83E-05	25

Unfortunately, these events cannot be easily addressed in primary differentiated adipocyte cultures. Firstly, the impact of chronic HFD and adipose tissue hypertrophy is a principal variable in the broadening of *Nr1d1*’s repressive influence. Secondly, as has been shown in the literature, NR1D1 is required for in vitro differentiation of pre-adipocytes into mature adipocytes (Kumar et al., 2010; Wang and Lazar, 2008).

4) The lack of specificity also becomes an issue with the analyses done in Figure 6 in which they have overlaid their RNA-Seq data with previously published ChIP-seq data. They then use bioinformatics to predict which genes are direct targets of REVERBa. This analysis would be strengthened if they first knew which of their transcriptional changes represented expression differences in adipocytes and if they then either did their own ChIP-seq study or at least validated some candidate genes by ChIP-PCR to actually confirm REVERBa binding.

As discussed above in response to points 1 and 3, we have now performed our own ChIP-seq study with the robust HaloNR1D1 system. The results of this study are described in the main manuscript on lines 258-300. We have used this new ChIP-seq data to examine the relationship between differentially regulated genes and sites of NR1D1 chromatin binding, and so infer the scope of NR1D1 action. We understand the reviewers/editors’ suggestion that we extend these studies with candidate genes. Therefore, we have conducted an additional study in which mature adipocytes were isolated from *Nr1d1^Flox2-6^:Adipoq^Cre^* and *Nr1d1^Flox2-6^* mouse under 16-week NC and HFD-feeding. Here, a set of candidate genes were chosen which (i) showed differential regulation in bulk adipose RNA-seq under HFD- but not NC-fed conditions; and (ii) were in close proximity to a HaloNR1D1 ChIP peak. Profiling gene expression in these adipocyte fractions confirmed both the absence of de-repression under the NC-feeding state, as well as an enhanced expression of target genes under HFD-feeding conditions in the *Nr1d1^Flox2-6^:Adipoq^Cre^* mice.

5) In Figure 3, the authors highlight that adipocyte-specific knockout of REVERBa results in increased collagen gene expression (Figure 3). However, on a high fat diet, mutant animals actually have decreased fibrosis. How do the authors explain this seeming contradiction? At a minimum, they should measure the same panel of collagen genes they propose to be regulated by REVERBa to see if they are altered in high fat fed gWAT.

We apologise for any lack of clarity in the original manuscript regarding the potential role of NR1D1 in regulating collagen and extracellular matrix dynamics. Our data strongly suggest that several collagen genes and collagen modifying enzymes are direct targets of NR1D1 repressive activity in gWAT. This is based on increased gene expression in the *Nr1d1^Flox2-6^:Adipoq^Cre^* mice compared to *Nr1d1^Flox2-6^* controlled (Figure 3F), as well as close proximity of de-repressed genes to NR1D1 ChIP peaks (for example, on simple nearest-gene annotation with HOMER, the genes *Col1a1, Col5a3, Col6a1, Col6a2, Col18a1, Loxl4* are found adjacent to NR1D1 peaks). These altered ECM dynamics may facilitate adipocyte/adipose tissue expansion under HFD-feeding conditions (discussed more below). However, this is likely to be quite a different process than the development of pronounced tissue fibrosis in the control animals under chronic HFD-feeding conditions.

The development of obesity-related WAT fibrosis is a complex process, with remodelling of the extracellular matrix (involving both collagen synthesis and breakdown) and contribution from both resident and infiltrating immune cells. Moreover, the fibrotic state of a tissue cannot be judged simply on collagen gene expression. The virtual absence of gWAT fibrosis in the HFD-fed mice lacking *Nr1d1* expression in adipocytes is most likely a reflection of an attenuated inflammatory response and reduced adipocyte dysfunction in comparison to obese control animals. Indeed, under a pathological obese state, a major contribution to collagen production, modification and fibril deposition would come from non-adipocyte cells (where NR1D1 activity is intact).

As suggested, we have measured the same panel of collagen genes in HFD gWAT (Author response image 4), and find that obesity does modulate observed patterns of gene expression. Even in this limited examination, some important differences remain (such as the reduced induction of *Col6a3* in response to chronic HFD-feeding in the adipocyte knockout mice). *Col6a3* has been associated with obesity, and obesity-related WAT inflammation and fibrosis in both animal models and humans (Khan et al., 2009; Pasarica et al., 2009; Sun et al., 2014).

**Author response image 4. respfig4:** Collagen gene expression in gWAT, as measured by qPCR. 2-way ANOVA with Tukey's multiple comparison tests. *P<0.05, **P<0.01.

Moreover, inspection of the RNA-seq data shows significant reduction in expression of numerous matrix metalloproteinase genes (e.g. *Mmp2, Mmp14, Mmp23, Mmp27*) (**Figure 5 Source Data File 1 – “Figure 5B_edgeR_HFD_Cre+vsCre-sigDN” tab**) in control HFD-fed samples compared to equivalent knockout samples, which suggests that ECM remodelling activity is different between the two genotypes. The same data show altered expression of fibulin and fibrillin genes between genotypes in the HFD state, indicating that the extracellular microfibrillar network is likely impacted as well.

Therefore, it is not contradictory to find that certain collagen genes are targets of NR1D1 repressive activity, but that *Nr1d1*-targeted adipose is less susceptible to fibrosis. Instead, the reduced fibrosis reflects differences in numerous processes, some of which are under direct NR1D1 control, and others which are indirectly affected by *Nr1d1* deletion. We have now added further comment on this subject in the manuscript text (lines 346-348).

6) Some of the claims made on the metabolic phenotyping are overstatements:a) The data presented on acute cold exposure and REVERBb compensation in 3T3-L1 adipocytes are not at all sufficient to claim that the phenotypes seen here are not due to effects in brown fat or due to compensatory action by REVERBb.

The key phenotypic finding revealed in our study is that loss of NR1D1 in adipocytes increases adipose tissue expansion in response to HFD-feeding. Therefore, it may be hypothesised that a decrease in energy expenditure and/or reduced thermogenesis could contribute to this phenotype. However, we and others have shown that daily mean energy expenditure and body temperature are not significantly altered in either the adipocyte-specific or the global *Nr1d1* knockout line (Delezie et al., 2012; Hand et al., 2015). Moreover, in our original manuscript, we demonstrate that daily body temperature profiles are not different between adipocyte-specific *Nr1d1* knockout mice under normal temperature, thermoneutral (~28-29°C), and cold-challenge conditions. These studies under different ambient temperature environments would be expected to reveal any major differences in inherent (thermoneutral) or stimulated brown adipose tissue (BAT) activity. We now also include assessment of metabolic gas exchange in *Nr1d1^Flox2-6^:Adipoq^Cre^* and *Nr1d1^Flox2-6^* mice, which similarly shows no change in energy expenditure between the two genotypes (Author response image 5). Taken together, these results strongly suggest that altered BAT activity does not significantly contribute to the adiposity phenotype of the mice under normal chow or chronic HFD feeding. Nevertheless, we have softened the relevant manuscript text (lines 142-146).

**Author response image 5. respfig5:** No change in metabolic parameters in Nr1d1^Flox2-6^:Adipoq^Cre^ mice. Representative two day recordings of metabolic gas exchange parameters of oxygen consumption (VO_2_), carbon dioxide production (VCO_2_), energy expenditure (kcal/hr) and respiratory exchange ratio (RER) in male mice. Average day and night readings from 5 days of recordings showed no alterations in metabolic parameters in Nr1d1^Flox2-6^:Adipoq^Cre^ mice compared to Nr1d1^Flox2-6^controls. Data shown as mean +/-SEM, or as individual values and mean. **P<0.01, indicating a difference due to time-of-day on 2-way ANOVA with Sidak’s multiple comparisons.

With regards to potential compensatory activity by NR1D2 (REVERBβ) in our *Nr1d1* knockout models, we acknowledge that it is difficult to completely rule out this possibility. Specifically, it is the paucity of phenotypic or transcriptional impact in the *Nr1d1^Flox2-6^:Adipoq^Cre^* mice under normal chow feeding conditions which raises a question of NR1D2 activity. However, several lines of evidence argue against this possibility: (i) expression of *Nr1d2* in WAT is not altered by the loss of *Nr1d1* in either the global or adipocyte-specific knockout mouse; (ii) if NR1D2 activity were sufficient to compensate for loss of *Nr1d1*, and in doing so mitigate transcriptional and/or phenotypic impact, then why is this not evident in the global *Nr1d1*^-/-^ mice (under normal chow conditions), or in the *Nr1d1^Flox2-6^:Adipoq^Cre^* mice under HFD-feeding conditions?; (iii) potential compensation by *Nr1d2* does not extend to transcriptional regulation of the core circadian clock genes, including *Arntl (Bmal1)* and *Nfil3*, which show significant de-repression upon *Nr1d1* deletion; (iv) finally, as we have shown in our in vitro studies (Figure 3 Supplemental), combined knockdown of both *Nr1d1* and *Nr1d2* in differentiated 3T3-L1 adipocytes does not cause an increase in the expression of putative metabolic targets. It is difficult to undertake similar studies in vivo. Dual deletion of *Nr1d1* and *Nr1d2* profoundly disrupts circadian clock function (core clock gene expression is rendered arrhythmic; e.g. Cho et al., 2012; Guan et al., 2020), and therefore, it is not possible to separate the impact of NR1D1/β loss from loss of circadian timing *per se* under these conditions.

Taken together, we do not believe that NR1D2 compensation is likely to play a major role in our models; nevertheless, we have softened the text in the main manuscript (lines 178-183).

b) Isolated sections with F4/80 alone are not sufficient to make a broad claim about inflammation. This would be more convincing if supported by flow cytometry data and inflammatory gene expression.

We appreciate that immunohistochemistry using a general macrophage marker has limitations with regards to describing overall immune response and inflammatory state within the adipose tissue of our different mice. Our aim was to broadly demonstrate that, despite showing enhanced obesity in response to HFD-feeding, mice lacking *Nr1d1* expression in adipocytes do not exhibit increased macrophage infiltration into gWAT (an extremely well-characterised pathological process associated with obesity in humans and animals). The reduced appearance of inflammatory cuffs was pronounced and consistent across our animals. The main focus of our current manuscript is the altered metabolic state and transcriptional regulation delivered by NR1D1 under normal conditions and during obesity, rather than the detailed immune consequence of enhanced lipid storage. Therefore, we have not undertaken flow cytometry-based immune cell characterisation. Nonetheless, to increased clarity and to reinforce the attenuated immune response observed in gWAT collected from HFD-fed *Nr1d1^Flox2-6^:Adipoq^Cre^* mice, we have more thoroughly analysed and presented immune-related gene expression within our RNA-seq studies. Using IPA (Qiagen), we assessed enrichment and overall direction of change in genes that showed differential expression between the two genotypes fed HFD for 16-weeks. Extensive enrichment of immune-related pathways were revealed by this analysis, and importantly, all of those showing significant enrichment were predicted to be strongly attenuated in the gWAT samples lacking *Nr1d1* expression. We have now also included these analyses within Figure 5 Source Data File 1 accompanying the main manuscript.

c) The legend for Figure 4 says that mutant mice have "greater insulin sensitivity". However, the curves in Figure 4F look very similar. Moreover, they calculate the area within the curve. I am not sure if they mean the area under the curve, which is more typically done, but this is confusing and needs to be clarified.

We apologise if the data has been presented in a confusing way. To show insulin sensitivity more clearly, we have plotted the data in Figure 4F as blood glucose change from baseline. Area under the curve is then calculated from this. We used the term ‘area within curve’ as negative effect of insulin on blood glucose levels entails that the area of interest lies above the curve, but agree that the term ‘area under the curve’ is more conventional.

References:

Cho, H., Zhao, X., Hatori, M., Yu, R.T., Barish, G.D., Lam, M.T., Chong, L.W., Ditacchio, L., Atkins, A.R., Glass, C.K., et al. (2012). Regulation of circadian behaviour and metabolism by REV-ERB-α and REV-ERB-β. Nature 485, 123–127.

Collins, J.M., Neville, M.J., Hoppa, M.B., and Frayn, K.N. (2010). de novo lipogenesis and stearoyl-CoA desaturase are coordinately regulated in the human adipocyte and protect against palmitate-induced cell injury. J. Biol. Chem. 285, 6044–6052.

Cunningham, P.S., Ahern, S.A., Smith, L.C., da Silva Santos, C.S., Wager, T.T., and Bechtold, D.A. (2016). Targeting of the circadian clock via CK1δ/ε to improve glucose homeostasis in obesity. Sci. Rep. 6, 29983.

Delezie, J., Dumont, S., Dardente, H., Oudart, H., Gréchez‐Cassiau, A., Klosen, P., Teboul, M., Delaunay, F., Pévet, P., and Challet, E. (2012). The nuclear receptor REV‐ERBα is required for the daily balance of carbohydrate and lipid metabolism. FASEB J. 26, 3321–3335.

Duez, H., van der Veen, J.N., Duhem, C., Pourcet, B., Touvier, T., Fontaine, C., Derudas, B., Baugé, E., Havinga, R., Bloks, V.W., et al. (2008). Regulation of Bile Acid Synthesis by the Nuclear Receptor Rev-erbα. Gastroenterology.

Dyar, K.A., Lutter, D., Artati, A., Ceglia, N.J., Liu, Y., Armenta, D., Jastroch, M., Schneider, S., de Mateo, S., Cervantes, M., et al. (2018). Atlas of Circadian Metabolism Reveals System-wide Coordination and Communication between Clocks. Cell 174, 1571-1585.e11.

Feng, D., Liu, T., Sun, Z., Bugge, A., Mullican, S.E., Alenghat, T., Liu, X.S., and Lazar, M.A. (2011). A circadian rhythm orchestrated by histone deacetylase 3 controls hepatic lipid metabolism. Science (80-. ). 331, 1315–1319.

Ferrari, A., Longo, R., Fiorino, E., Silva, R., Mitro, N., Cermenati, G., Gilardi, F., Desvergne, B., Andolfo, A., Magagnotti, C., et al. (2017). HDAC3 is a molecular brake of the metabolic switch supporting white adipose tissue browning. Nat. Commun.

Forman, B.M., Chen, J., Blumberg, B., Kliewer, S.A., Henshaw, R., Ong, E.S., and Evans, R.M. (1994). Cross-talk among ROR α 1 and the Rev-erb family of orphan nuclear receptors. Mol. Endocrinol. 8, 1253–1261.

Guan, D., Xiong, Y., Trinh, T.M., Xiao, Y., Hu, W., Jiang, C., Dierickx, P., Jang, C., Rabinowitz, J.D., and Lazar, M.A. (2020). The hepatocyte clock and feeding control chronophysiology of multiple liver cell types. Science (80-. ). 369, 1388–1395.

Hand, L.E., Usan, P., Cooper, G.J.S., Xu, L.Y., Ammori, B., Cunningham, P.S., Aghamohammadzadeh, R., Soran, H., Greenstein, A., Loudon, A.S.I., et al. (2015). Adiponectin Induces A20 Expression in Adipose Tissue to Confer Metabolic Benefit. Diabetes 64, 128–136.

Hunter, A.L., Pelekanou, C.E., Adamson, A., Downton, P., Barron, N.J., Cornfield, T., Poolman, T.M., Humphreys, N., Cunningham, P.S., Hodson, L., et al. (2020). Nuclear receptor NR1D1 is a state-dependent regulator of liver energy metabolism. Proc. Natl. Acad. Sci. 117, 25869–25879.

Khan, T., Muise, E.S., Iyengar, P., Wang, Z. V., Chandalia, M., Abate, N., Zhang, B.B., Bonaldo, P., Chua, S., and Scherer, P.E. (2009). Metabolic Dysregulation and Adipose Tissue Fibrosis: Role of Collagen VI. Mol. Cell. Biol. 29, 1575–1591.

Kumar, N., Solt, L.A., Wang, Y., Rogers, P.M., Bhattacharyya, G., Kamenecka, T.M., Stayrook, K.R., Crumbley, C., Floyd, Z.E., Gimble, J.M., et al. (2010). Regulation of adipogenesis by natural and synthetic REV-ERB ligands. Endocrinology 151, 3015–3025.

Li, P., Fan, W., Xu, J., Lu, M., Yamamoto, H., Auwerx, J., Sears, D.D., Talukdar, S., Oh, D., Chen, A., et al. (2011). Adipocyte NCoR knockout decreases PPARγ phosphorylation and enhances PPARγ activity and insulin sensitivity. Cell.

Nguyen, H.C.B., Adlanmerini, M., Hauck, A.K., and Lazar, M.A. (2020). Dichotomous engagement of HDAC3 activity governs inflammatory responses. Nature.

Pasarica, M., Gowronska-Kozak, B., Burk, D., Remedios, I., Hymel, D., Gimble, J., Ravussin, E., Bray, G.A., and Smith, S.R. (2009). Adipose tissue collagen VI in obesity. J. Clin. Endocrinol. Metab. 94, 5155–5162.

Phelan, C.A., Gampe, R.T., Lambert, M.H., Parks, D.J., Montana, V., Bynum, J., Broderick, T.M., Hu, X., Williams, S.P., Nolte, R.T., et al. (2010). Structure of Rev-erbα bound to N-CoR reveals a unique mechanism of nuclear receptor–co-repressor interaction. Nat. Struct. Mol. Biol. 17, 808–814.

Preitner, N., Damiola, F., Luis-Lopez-Molina, Zakany, J., Duboule, D., Albrecht, U., and Schibler, U. (2002). The orphan nuclear receptor REV-ERBα controls circadian transcription within the positive limb of the mammalian circadian oscillator. Cell 110, 251–260.

Quagliarini, F., Mir, A.A., Balazs, K., Wierer, M., Dyar, K.A., Jouffe, C., Makris, K., Hawe, J., Heinig, M., Filipp, F.V., et al. (2019). Cistromic Reprogramming of the Diurnal Glucocorticoid Hormone Response by High-Fat Diet. Mol. Cell 76, 531-545.e5.

Roh, H.C., Kumari, M., Taleb, S., Tenen, D., Jacobs, C., Lyubetskaya, A., Tsai, L.T.Y., and Rosen, E.D. (2020). Adipocytes fail to maintain cellular identity during obesity due to reduced PPARγ activity and elevated TGFβ-SMAD signaling. Mol. Metab.

Sun, K., Park, J., Gupta, O.T., Holland, W.L., Auerbach, P., Zhang, N., Goncalves Marangoni, R., Nicoloro, S.M., Czech, M.P., Varga, J., et al. (2014). Endotrophin triggers adipose tissue fibrosis and metabolic dysfunction. Nat. Commun. 5, 3485.

Wang, J., and Lazar, M.A. (2008). Bifunctional Role of Rev-erbα in Adipocyte Differentiation. Mol. Cell. Biol. 28, 2213–2220.

Yin, L., and Lazar, M.A. (2005). The orphan nuclear receptor Rev-erbalpha recruits the N-CoR/histone deacetylase 3 corepressor to regulate the circadian Bmal1 gene. Mol. Endocrinol. 19, 1452–1459.

Zamir, I., Harding, H.P., Atkins, G.B., Hörlein, A., Glass, C.K., Rosenfeld, M.G., and Lazar, M.A. (1996). A nuclear hormone receptor corepressor mediates transcriptional silencing by receptors with distinct repression domains. Mol. Cell. Biol. 16, 5458–5465.

Zamir, I., Zhang, J., and Lazar, M.A. (1997). Stoichiometric and steric principles governing repression by nuclear hormone receptors. Genes Dev. 11, 835–846.

Zhang, Y., Fang, B., Emmett, M.J., Damle, M., Sun, Z., Feng, D., Armour, S.M., Remsberg, J.R., Jager, J., Soccio, R.E., et al. (2015). Discrete functions of nuclear receptor Rev-erbalpha couple metabolism to the clock. Science (80-. ). 348, 1488–1492.

Zhang, Y., Dallner, O.S., Nakadai, T., Fayzikhodjaeva, G., Lu, Y.H., Lazar, M.A., Roeder, R.G., and Friedman, J.M. (2018). A noncanonical PPARγ/RXRα-binding sequence regulates leptin expression in response to changes in adipose tissue mass. Proc. Natl. Acad. Sci. U. S. A. 115, E6039–E6047.